# Time-resolved in vivo ubiquitinome profiling by DIA-MS reveals USP7 targets on a proteome-wide scale

Martin Steger [1,5✉], Vadim Demichev[2,3,4,5], Mattias Backman[1], Uli Ohmayer [1], Phillip Ihmor[1], Stefan Müller[1], Markus Ralser [2,3] & Henrik Daub[1]

Mass spectrometry (MS)-based ubiquitinomics provides system-level understanding of ubiquitin signaling. Here we present a scalable workflow for deep and precise in vivo ubiquitinome profiling, coupling an improved sample preparation protocol with data-independent acquisition (DIA)-MS and neural network-based data processing specifically optimized for ubiquitinomics. Compared to data-dependent acquisition (DDA), our method more than triples identification numbers to 70,000 ubiquitinated peptides in single MS runs, while significantly improving robustness and quantification precision. Upon inhibition of the oncology target USP7, we simultaneously record ubiquitination and consequent changes in abundance of more than 8,000 proteins at high temporal resolution. While ubiquitination of hundreds of proteins increases within minutes of USP7 inhibition, we find that only a small fraction of those are ever degraded, thereby dissecting the scope of USP7 action. Our method enables rapid mode-of-action profiling of candidate drugs targeting DUBs or ubiquitin ligases at high precision and throughput.

[1] Evotec München GmbH, Am Klopferspitz 19a, Martinsried, Germany. [2] Charité – Universitätsmedizin Berlin, Department of Biochemistry, Berlin, Germany. [3] The Francis Crick Institute, Molecular Biology of Metabolism Laboratory, London, UK. [4] The University of Cambridge, Department of Biochemistry and Cambridge Centre for Proteomics, Cambridge, UK. [5] These authors contributed equally: Martin Steger, Vadim Demichev. ✉email: martin.steger@evotec.com

The ubiquitin-proteasome system (UPS) consists of numerous proteins, including ubiquitin ligases, ubiquitin proteases (also termed deubiquitinases or DUBs) and the proteasome. Approximately 750 enzymes mediate ubiquitin attachment to and its cleavage from target proteins and regulate a myriad of intracellular processes, such as cell cycle progression, selective autophagy, or the response to growth factors[1]. Consequently, dysregulation of the UPS can contribute to loss of cell cycle control and ultimately to carcinogenesis[2]. On the other hand, different components of the UPS are targets for anticancer drugs. For example, many proteasome inhibitors and E3 ligase modulators are clinically approved, the latter ones being of high interest because they can be exploited for degrading proteins otherwise considered undruggable[3,4].

Ubiquitin is attached via its C-terminal carboxyl group to lysine (K) side chains and protein N-termini to produce a monoubiquitinated target protein. As each ubiquitin molecule contains eight ubiquitin attachment sites (seven K residues within and one at its amino terminus), consecutive linkage of additional ubiquitin monomers leads to the formation of polymeric chains. Polyubiquitin chains can form on each of these K residues and encode for specific signals. For example, while K11- and K48-linked chains can tag proteins for proteasomal degradation, attachment of K63-linked ubiquitin conjugates to target proteins tends to modulate protein-protein interactions[5].

While early studies on ubiquitinated proteins were conducted on a target-by-target basis, mass spectrometry (MS)-based proteomics has facilitated global ubiquitin signaling profiling, such as B-cell receptor signaling[6,7]. The primary method for ubiquitinome analyses relies on immunoaffinity purification and MS-based detection of diglycine-modified peptides (K-GG), generated by tryptic digestion of ubiquitin-modified proteins (referred to as ubiquitinomics or ubiquitylomics)[8–11].

Here, we report a scalable and robust workflow for mass spectrometry-based ubiquitinomics. By introducing a sodium deoxycholate (SDC)-based lysis protocol, and exploiting data-independent acquisition mass spectrometry (DIA-MS) coupled to deep neural network-based data processing, we boost reproducibility, identification numbers, and quantitative accuracy. We demonstrate the power of our method by comprehensively mapping substrates of the deubiquitinase USP7, an actively investigated anticancer drug target shown to regulate the tumor suppressor p53[12,13]. Following inhibition with selective inhibitors, we profile the dynamics of both the proteome and the ubiquitinome at high temporal resolution. Combining the profiles of ubiquitinated peptides with their corresponding protein abundances not only allows us to define putative USP7 targets with high confidence, but also to distinguish regulatory ubiquitination leading to protein degradation from non-degradative events.

## Results

### Optimized cell lysis protocol for MS-based ubiquitinomics.
For improving depth and precision of ubiquitin-remnant peptide quantification by mass spectrometry (MS), we optimized sample preparation, MS, and data analysis. Despite its good performance in regular proteomic analyses[14,15], sodium deoxycholate (SDC)-based protein extraction has only recently been explored for ubiquitinomics applications[16]. We devised a modified version of this lysis protocol, by supplementing the SDC buffer with chloroacetamide (CAA). We reasoned that immediate boiling of samples after lysis together with the high concentration of CAA would increase ubiquitin site coverage, as CAA rapidly inactivates cysteine ubiquitin proteases by alkylation. In contrast to the less reactive CAA, it has been reported that iodoacetamide can cause di-carbamidomethylation of lysine residues, which mimic

ubiquitin remnant K-GG peptides in terms of mass tag added (both 114.0249 Da)[17]. We confirmed that CAA does not induce any unspecific di-carbamidomethylation of lysine residues, even when incubated at high temperatures (Supplementary Fig. 1a). To directly compare SDC with the conventional urea-based lysis buffer[18], we treated HCT116 cells with the proteasome inhibitor MG-132 for 6 h and extracted proteins with either buffer. After tryptic digestion of proteins and immunoaffinity purification of K-GG remnant peptides, we acquired the data in data-dependent acquisition (DDA) mode. This led to the identification of 41,094 K-GG remnant peptides and we found on average 6 –GG modified lysine residues per protein. SDC-based lysis yielded on average 38% more K-GG peptides than urea buffer (26,756 vs 19,403, $n = 4$, workflow replicates), without negatively affecting the relative enrichment specificity (Fig. 1a, b, Supplementary Fig. 1b and Supplementary Data 1). Moreover, SDC increased both the number of precisely quantified K-GG peptides (i.e., peptides with a coefficient of variation (CV) < 20%) and the reproducibility (Supplementary Fig. 1c, d).

In parallel, we digested different amounts of MG-132-treated Jurkat cell lysate (31 μg–4 mg protein, $n = 4$ for each condition) and enriched K-GG peptides before MS analysis. Here we also quantified about 30,000 K-GG peptides from 2 mg of protein input, with identification numbers dropping below 20,000 for inputs of 500 μg or less (Fig. 1c, Supplementary Fig. 1e and Supplementary Data 2). Next, we benchmarked our SDC-based lysis protocol against the recently described UbiSite method that relies on urea lysis and immunoaffinity purification of K-GGRLRLVLHLTSE remnant peptides resulting from Lys-C digestion of ubiquitinated proteins[19]. We processed three of our single-shot Jurkat samples (2 mg protein input) together with three biological replicates (16 high pH reversed-phase fractions each) of proteasomal inhibitor-treated Jurkat cells from Akimov et al in MaxQuant[20]. UbiSite quantified on average 30% more K-GG peptides in the three biological replicate samples. However, our single-shot SDC workflow yielded a higher number of precisely quantified peptides (CV < 20%) and we obtained a much better enrichment specificity. Moreover, our protocol required 20-times less protein input, and only 1/10th of the MS acquisition time per sample. We hence concluded that for a majority of applications, the SDC-based lysis protocol would be advantageous (Fig. 1d, e and Supplementary Data 3).

### DIA-MS and neural network-based data processing boost ubiquitinome coverage.
Even though our revised lysis protocol quantified up to 30,000 K-GG peptides in 125 min LC-MS runs, due to the semi-stochastic sampling inherent to DDA, only about 50% of those identifications were without missing values in replicate samples (Supplementary Fig. 1d). Consequently, the number of robustly quantified K-GG peptides would be greatly reduced in large sample series. To address this problem, we explored data-independent acquisition (DIA), an acquisition technique that is less susceptible to run-to-run variability[21]. Being actively developed, DIA allows the identification and quantification of very high numbers of peptides and proteins, for either regular proteomes or phosphoproteomes[22,23]. Recently we introduced DIA-NN, a deep neural network-based software, which significantly increases proteomic depth and quantitative accuracy for DIA, especially for samples of high complexity[24]. To improve the analysis of DIA data for ubiquitinomics, we expanded DIA-NN with an additional scoring module that ensures confident identification of modified peptides, including K-GG peptides (Methods).

Using a medium-length (75 min) nanoLC gradient, we set up optimized MS methods (Supplementary Data 4) and benchmarked

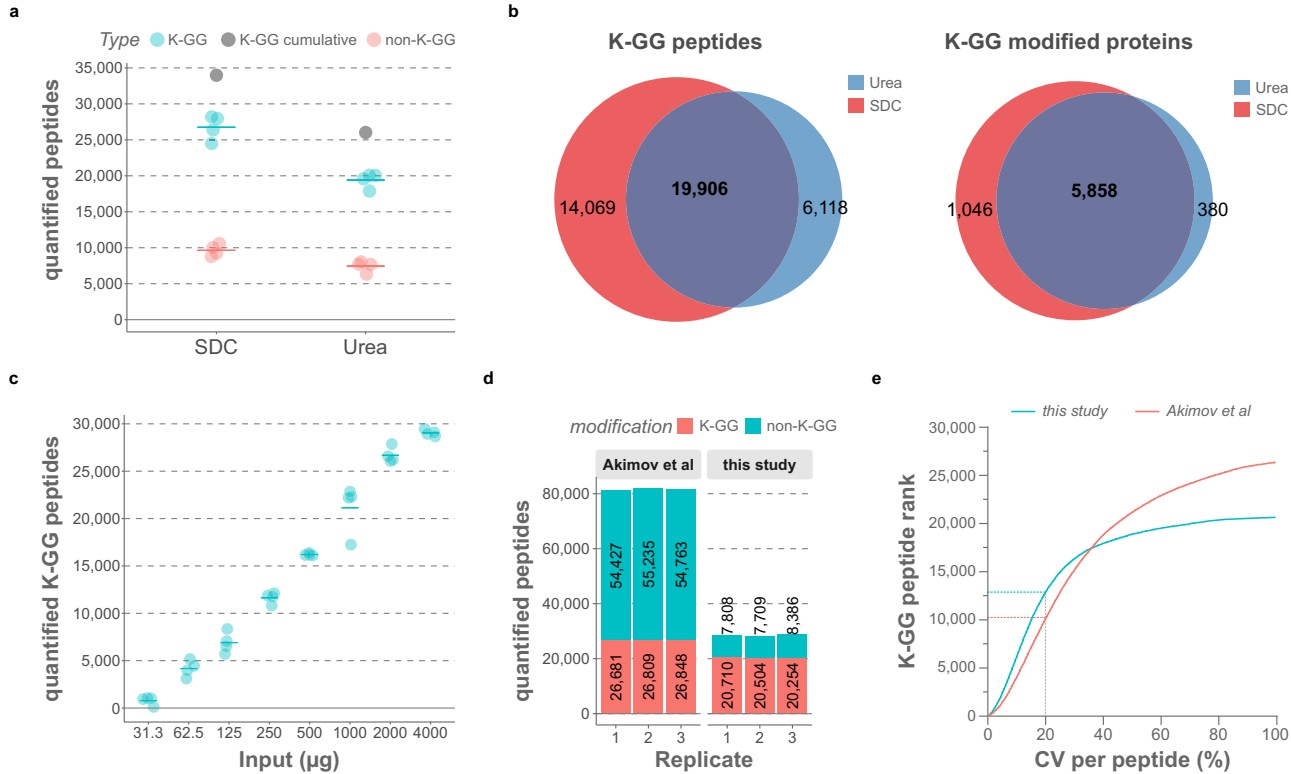

**Fig. 1 Comparison of urea- and SDC-based cell lysis for mass spectrometry-based ubiquitinomics. a** Fraction of unmodified and K-GG modified peptides quantified from either urea or SDC lysates in MG-132-treated HCT116 cells. Four individual samples were processed for each lysis protocol, with 2 mg of protein input per replicate. Only half of each sample was injected into the MS, which was operated in DDA mode (125 min LC gradient). The raw data were processed with MaxQuant, with "match between runs" (MBR) enabled. Gray dots show cumulative numbers of K-GG peptide identifications from four replicates. **b** Overlap of quantified K-GG peptides and K-GG modified proteins with urea and SDC lysis buffers. **c** MS-quantified ubiquitinated peptides (K-GG remnants) with different protein inputs in MG-132-treated Jurkat cells (6 h). Four individual samples were processed for each condition and the data were acquired in single-shot mode using a 125 min DDA-MS method. The raw files were processed with MaxQuant, with match-between-runs (MBR) activated. **d** Fraction of unmodified and K-GG modified peptides quantified with the SDC-based lysis protocol and with UbiSite[19]. Three single-shot runs of enriched K-GG peptides from MG-132-treated Jurkat cells (2 mg of input, three replicates of samples as shown in **c**) were processed with MaxQuant (with MBR) together with three biological replicates of bortezomib-treated Jurkat samples (16 high pH-reversed phase fractions for each replicate) from Akimov et al. **e** Ranked K-GG peptide coefficients of variation (CVs) for samples shown in **d**. The 20% CV cut-off is marked. Source data are provided as a Source Data file.

our DIA workflow against state-of-the-art label-free DDA. For DIA data processing we used DIA-NN in "library-free mode" (that is searching against a sequence database without an experimentally-generated spectral library), whereas for DDA data processing we used MaxQuant[20], with match-between-runs (MBR) enabled (Fig. 2a). While DDA quantified 21,434 K-GG peptides on average per sample from proteasome inhibitor-treated HCT116 cells, DIA more than tripled this number, to 68,429 K-GG peptides (Fig. 2b). Besides increased coverage, DIA showed excellent quantitative precision and reproducibility. The median CV for all quantified K-GG peptides was about 10%, and 68,057 peptides were quantified in at least three replicates (Fig. 2c, d). 88% of ubiquitinated peptides detected by DDA were also identified by DIA (Fig. 2e).

Although DIA-NN uses a rigorous approach to determine the false discovery rate (FDR)[24,25], it has not been validated for the identification of K-GG peptides. We therefore determined the identification confidence specifically for K-GG peptides experimentally, and showed that it is comparable to that of the DDA workflow and to a different DIA processing software (Supplementary Note 1 and Supplementary Fig. 10). We also confirmed the excellent quantitative accuracy and dynamic range of our DIA workflow, using a pool of synthetic K-GG peptides spiked into a yeast tryptic digest at different concentrations (Supplementary Note 2 and Supplementary Fig. 11).

We next benchmarked the performance of DIA-NN when using it in combination with an ultra-deep spectral library generated by high-pH reversed-phase fractionation, consisting of 146,626 K-GG peptides (Methods). This yielded similar results as library-free analysis, both in terms of coverage and reproducibility (Fig. 2b–f). Finally, to directly compare DIA-NN's performance for analyzing ubiquitinomics data to another software, we processed six raw files from Hansen et al.[16] with DIA-NN, revealing that DIA-NN identified on average 40% more K-GG peptides (Fig. 2g).

**DIA-MS ubiquitinomics for low protein amounts and high-throughput analyses**. When studying intracellular protein ubiquitination by mass spectrometry, proteasome inhibitors are often used to prevent degradation of the target of interest, thus conserving and boosting the ubiquitin signal[9,19,26]. However, such compounds are highly cytotoxic and result in an accumulation of newly synthesized, misfolded proteins, thus activating the unfolded protein response, one of the major cellular anti-stress mechanisms[27]. Moreover, proteasome inhibition globally perturbs protein turnover by disconnecting protein ubiquitination from degradation[8]. To determine whether our DIA-MS method is suitable for high coverage ubiquitinomics in the physiological context and in case of low protein input, we quantified K-GG peptides derived from different protein input

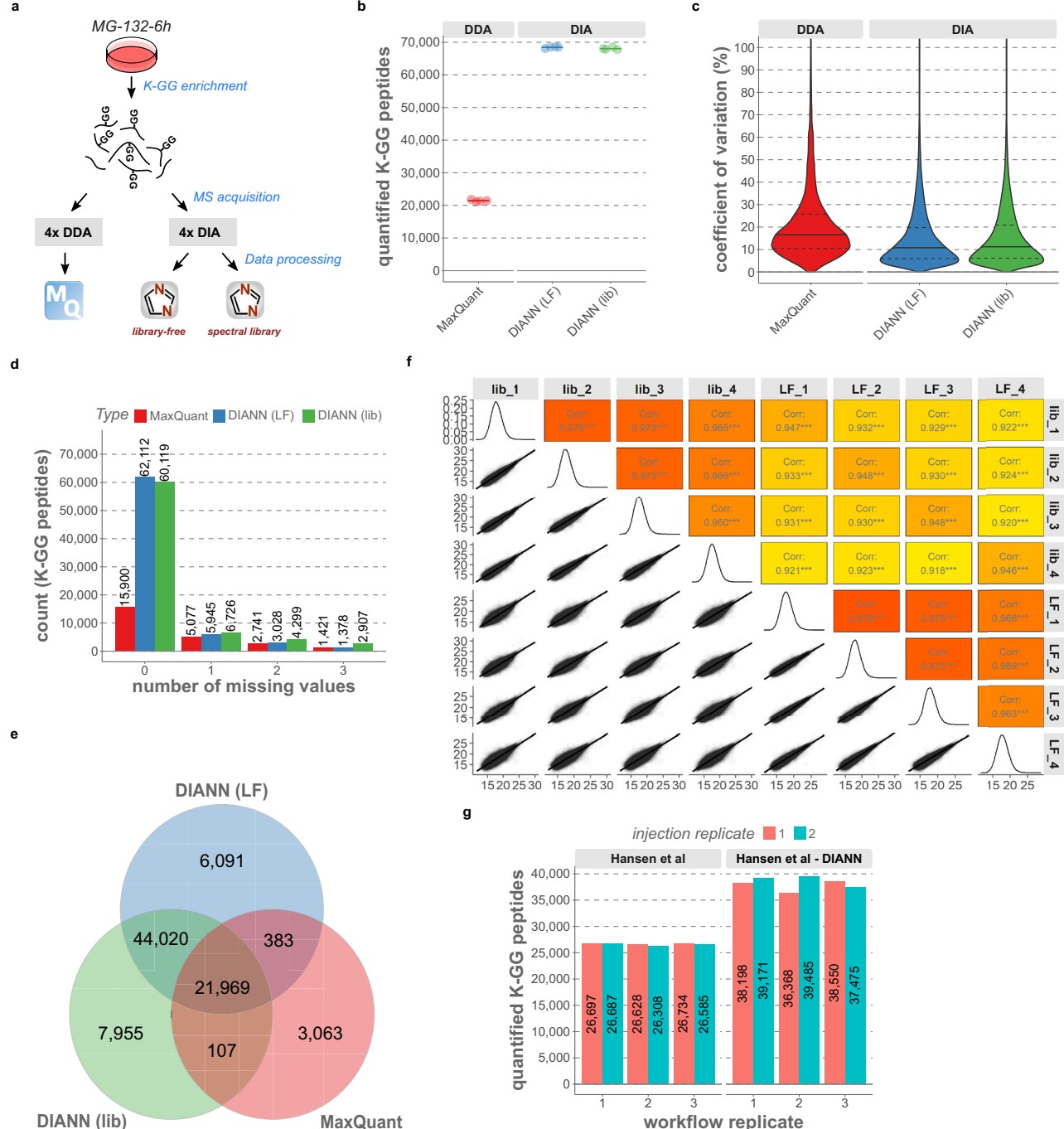

**Fig. 2 Side-by-side comparison of DDA and DIA for ubiquitinomics. a** Schematic of data-dependent acquisition (DDA) and data-independent acquisition (DIA) strategies. A K-GG peptide sample was prepared from MG-132-treated HCT116 cells and four replicates were acquired with each scan mode. The DDA data were processed with MaxQuant (MBR enabled). For DIA data processing in DIA-NN, the same DIA-MS raw files were processed with either library-free (no experimental spectral library) or library-based search (experimental spectral library obtained through high pH reversed-phase fractionation of peptides prior to K-GG peptide enrichment). **b** Quantified K-GG peptides with 2 mg of total protein as input for each replicate, in a side-by-side comparison of DDA and DIA-MS. Four samples (injection replicates) were acquired with 75 min single-shot LC-MS runs and processed with MaxQuant or DIA-NN. Library-based DIA data analysis was performed with an ultra-deep K-GG peptide library generated by library-free DIA analysis of high-pH fractions, as detailed in "Methods" (LF = library-free, lib = experimentally-generated spectral library). **c** Coefficients of variation (CVs, in %) for K-GG peptides as shown in **b** are plotted for each method. Continuous lines demarcate the median and dashed lines the upper and lower quartiles. **d** Number of identifications with 0, 1, 2, or 3 missing values in four replicates (samples shown in **b**) for DDA and DIA. **e** Venn diagram showing overlap of ubiquitinated peptides (K-GG) identified by DDA and by DIA (DIA-NN's library-free (LF) and spectral library-based (lib) approaches). **f** Scatter plot of Log2-transformed K-GG peptide intensities obtained from library-free and spectral library-based search options in DIA-NN (n = 4, injection replicates). Values indicate Pearson correlation coefficients. **g** Raw data of MG-132-treated HEK293 cells from Hansen et al. (Fig. 1d in the original manuscript, three workflow replicates, each injected in duplicate and processed with Spectronaut's directDIA mode[16]) were processed with DIA-NN (library-free mode). Plots show identified K-GG peptides (unique peptide sequences) for both Software packages. Source data are provided as a Source Data file.

amounts of either MG-132- or DMSO-treated cells. This revealed that only 1 mg of total protein was sufficient to reach saturation of about 70,000 quantified K-GG peptides in MG-132-treated cells. In DMSO-treated cells, however, increasing the input protein amount to 2 mg augmented the number of quantified K-GG remnants from 39,000 to 46,000 (Fig. 3a and Supplementary Data 5). Notably, with just 250 µg of protein, we quantified about 18,000 K-GG peptides in the absence of proteasome inhibition, demonstrating the feasibility of in-depth ubiquitinomics in the physiological context, even for low protein amounts. As expected, a large fraction of K-GG peptides increased in intensity upon proteasome inhibition and about 20,000 peptides increased >4-fold. On the other hand, about 1,500 and 6,200 K-GG peptides decreased in intensity >4-fold and >2-fold, respectively. Among those were K-GG sites mapping to the C-terminus of the histones H2A and H2B (K119 and K121, respectively), both of which are well-studied ubiquitination sites involved in the regulation of transcription and in DNA repair[28,29]. Besides these known sites, we also identified uncharacterized K-GG sites on histones that were strongly reduced in intensity upon MG-132 treatment, such as H2B-K44 (Fig. 3b).

To evaluate the potential for higher throughput applications, we set up DIA-MS methods for 15, 30, 45, and 60 min LC gradients (Supplementary Data 4), and measured K-GG-enriched HCT116 samples obtained from 2 mg of protein input. This showed that identification numbers improve massively from 15 to 30 min and from 30 to 45 min LC gradient length (+81% and +31%, respectively), while the gain in K-GG peptide IDs was moderate when further increasing MS acquisition time (8%, for 45–60 min and 6%, for 60–75 min). Even the 15 min LC-MS runs consistently quantified more than 20,000 K-GG peptides with good precision, demonstrating the potential for high-throughput ubiquitinomics applications (Fig. 3c, d and Supplementary Data 6). To benchmark DIA-NN's performance for analyzing samples not treated with proteasome inhibitors, we processed six raw files (DMSO-treated) from Hansen et al in library-free mode. Also in this comparison, DIA-NN clearly outperformed another DIA data processing software, quantifying on average almost 75% more K-GG peptides (Fig. 3e).

**In-depth quantitative ubiquitin site profiling for USP7 substrate identification**. Currently, there are six clinically approved drugs targeting different components of the UPS and several more in preclinical testing[4]. The unbiased analysis of ubiquitination and deubiquitination events by MS-based ubiquitinomics represents a powerful approach to develop or challenge a drug-targeting hypothesis. However, key aspects such as the modulation and the time-resolved interplay of protein ubiquitination and degradation upon drug treatment have been difficult to address on a truly proteome-wide scale. To meet this challenge, we sought to profile the selective USP7 inhibitor FT671[30] with our DIA workflow in a series of time-course experiments. The deubiquitinase USP7 is a potential anticancer drug target that is of particular interest for the treatment of hematological malignancies[12,13,31]. For example, USP7 deubiquitinates and stabilizes the E3 ligase Mdm2. As Mdm2 promotes p53 degradation, inhibition of USP7 leads to p53-dependent tumor growth suppression[32,33]. However, recent reports indicate that the ubiquitin signaling network regulated by USP7 may be much more complex than initially thought[30,33–35]. In this context, global proteomics approaches connecting early protein ubiquitination with later changes in protein abundance represent an elegant tool for pinpointing putative USP7 substrates with high confidence.

First, we quantified the ubiquitinomes and the proteomes at five individual time points after USP7 inhibition with FT671 and in the absence of proteasome inhibitors, to a depth of 45,000

K-GG peptides and over 10,000 proteins in each sample (Fig. 4a, Supplementary Fig. 2 and Supplementary Data 7). Principal component analysis (PCA) of the ubiquitinome captured the variance of the different treatment times and revealed a clear separation of DMSO and FT671 treatments at each time point.

We mapped at least one ubiquitination site on 8,087 proteins. Already after 15 min of USP7 inhibition, we found 1,243 ubiquitinated peptides (mapping to 552 proteins) that were significantly upregulated by more than twofold (Supplementary Fig. 3). However, only 42 of these rapidly ubiquitinated proteins were significantly downregulated by more than 20% over the 6 h time course (Supplementary Data 7). These 42 factors most likely represent direct USP7 targets marked for proteasomal degradation, and gene annotation enrichment analysis revealed that they were overrepresented in transcriptional regulation (Zinc finger) and the ubiquitin conjugation pathway (Fig. 4b, c and Supplementary Data 8). We detected upregulated ubiquitination sites for many reported USP7 targets, such as Trim27, UvssA, or Mdm2[32,34,35]. Besides these, we identified protein ubiquitination that was followed by degradation for a number of proteins without any reported connection to USP7 (e.g., Rnf220, L3mbtl2, Pcgf6). However, the majority of ubiquitinated proteins, such as USP7 itself or the USP7 interaction partner Daxx[36], were not degraded, even though multiple sites were strongly upregulated throughout the time course (Supplementary Fig. 4). Our time-resolved quantification of both the proteome and the ubiquitinome in the absence of proteasome inhibition thus distinguishes ubiquitination triggering apparent protein degradation from nondegradative ubiquitin signatures.

Although five Mdm2 sites were significantly upregulated already after 15 min, its total protein level only transiently decreased to return to baseline thereafter (Fig. 4d). This most likely resulted from a p53-controlled increase of Mdm2 transcription, as expected and reported previously[33]. In fact, ubiquitination sites on p53 initially decreased in intensity, resulting in protein stabilization at later time points, which coincided with an increased expression of its effector p21 (CDKN1A). Interestingly, besides these tumor suppressors, the proto-oncogenes c-Fos and c-Jun were both upregulated upon USP7 inhibition (Fig. 4d).

To confirm that the observed increase in protein ubiquitination is USP7-dependent, we knocked down USP7 expression using two different siRNA pools, treated the cells with FT671 for 5 min and extracted the proteins for ubiquitinome analysis. Again, we quantified over 40,000 K-GG peptides and close to 10,000 proteins in each sample with very good precision and MS-based quantification and Western blotting showed a fourfold reduction in USP7 expression upon knockdown (Supplementary Fig. 5a–f). While the compound treatment alone did not significantly alter the expression of any of the detected proteins at 5 min of treatment time, knockdown of USP7 induced profound changes in protein expression, including a stabilization of p53 (Supplementary Fig. 5g, h). As expected, we found that upon USP7 knockdown, for multiple targets identified in our earlier time course experiment the measured protein ubiquitination after FT671 treatment was strongly attenuated. This confirmed that FT671 is selectively targeting USP7 and further validated multiple previously unknown putative USP7 substrates (Fig. 4e and Supplementary Data 9).

To further increase the confidence of defining primary USP7 targets and to get more insights into USP7-mediated signaling, we mapped our ubiquitinomics data onto a USP7 interaction network retrieved from BioGrid[37]. We identified significantly upregulated sites at 15 min of FT671 treatment on 63% of high-confidence physical USP7 interactors (evidence from four experiments), strengthening the evidence that these proteins likely represent direct USP7 targets (Fig. 4f).

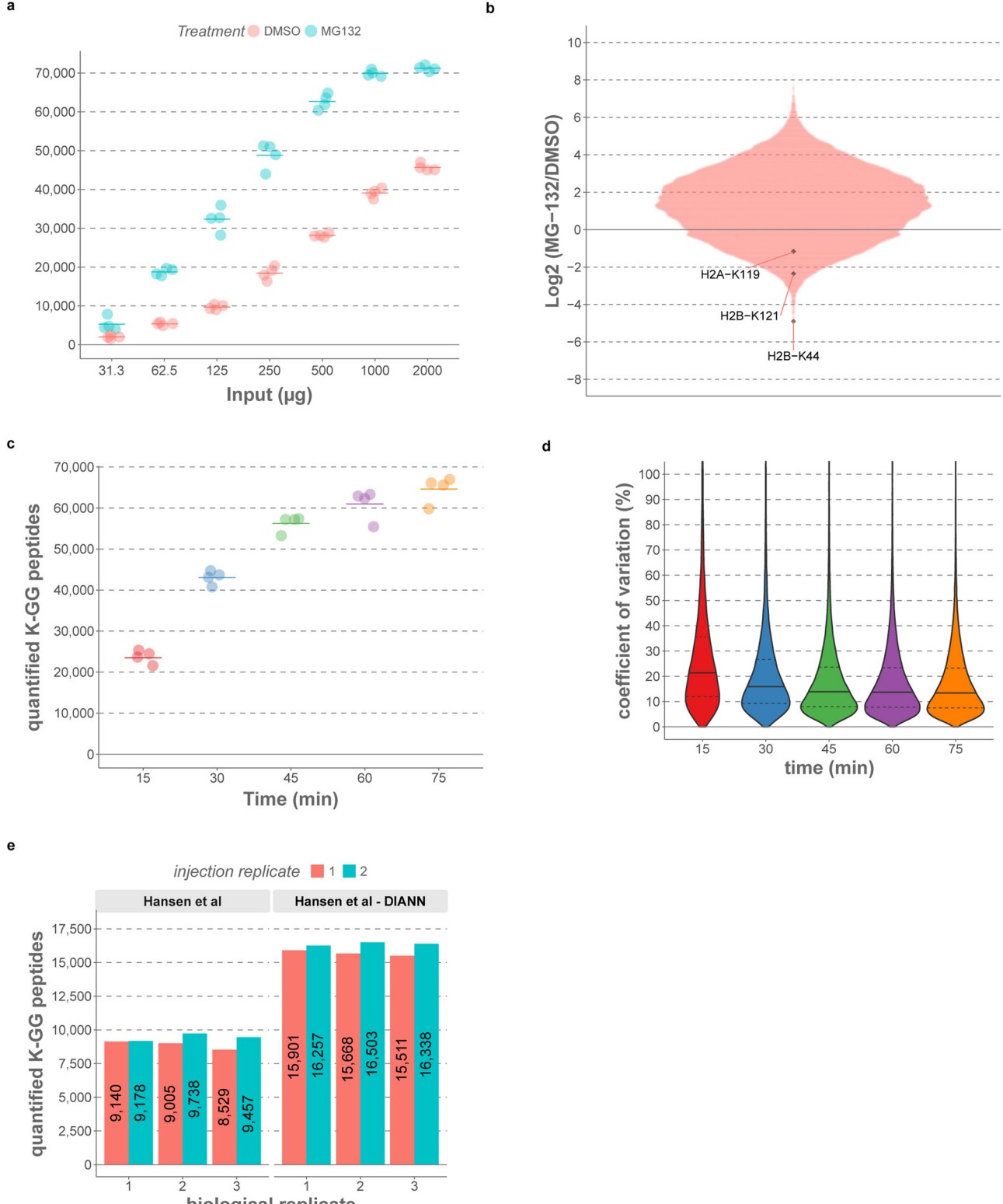

**Fig. 3 DIA-MS ubiquitinomics for untreated samples and for high-throughput applications. a** HCT116 cells were treated with DMSO or 10 μM MG-132 for 6 h and K-GG peptides from different protein inputs quantified by 75 min single-run DIA-MS (library-free mode). Four samples were processed independently for each tested protein input. **b** Dot plot of Log2-transformed K-GG peptide ratios (MG-132 vs DMSO) with 2 mg of protein input, as shown in **a**. Selected K-GG sites are highlighted. **c** Number of quantified K-GG peptides with 15, 30, 45, 60, and 75 min DIA-MS runs. A K-GG peptide pool was injected on the same instrument for each method (*n* = 4, injection replicates). The data were processed with library-free DIA. **d** Coefficients of variation (CVs) distributions for K-GG peptides quantified in **c**. Continuous lines in the violin plots demarcate the median and dashed lines upper and lower quartiles. **e** Shown are K-GG peptide numbers from DMSO-treated U2OS cells processed by Hansen et al. (Fig. 3 in the original manuscript, three biological replicates, each injected twice[16]) and the same raw files processed with DIA-NN (library-free mode). Source data are provided as a Source Data file.

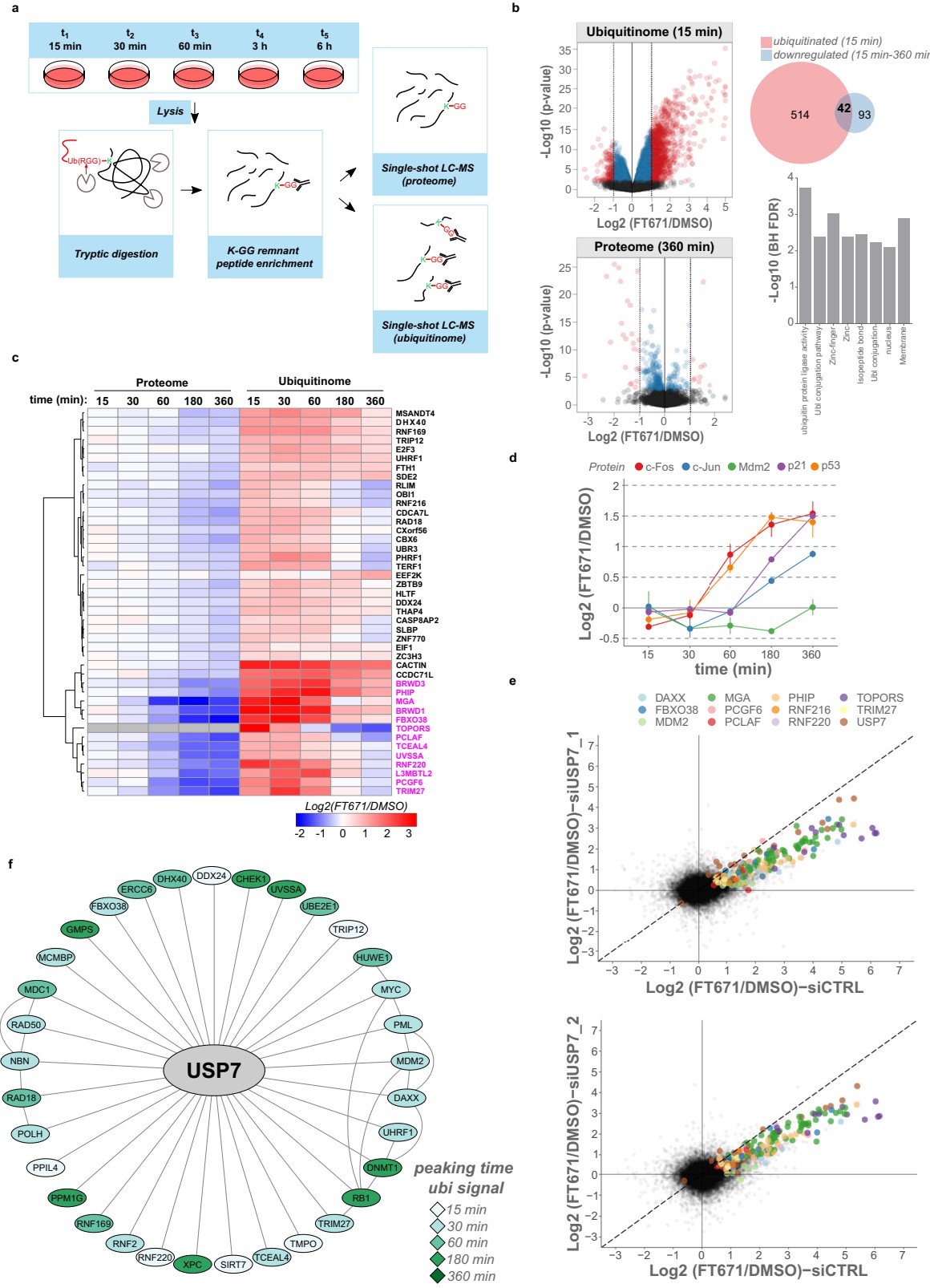

The strikingly high regulation of protein ubiquitination already at 15 min of USP7 inhibition pointed to a rapidly regulated, complex signaling network. To further dissect which of the upregulated ubiquitination sites are directly controlled by USP7, we performed a high-resolution time-course experiment with much faster sampling (at 2, 6 and 10 min), again quantifying both the ubiquitinome and the proteome in single-shot mode. At these early time points, we did not observe significant protein changes except for the downregulation of the E3 SUMO/ubiquitin ligase Topors at 10 min (Supplementary Fig. 6a). Topors was strongly degraded also in our initial time-course experiment and was below detection limit already 15 min after USP7 inhibition (Fig. 4c). In contrast, even at 2 min of FT671 treatment, we found 927 ubiquitinated peptides that were significantly and more than

**Fig. 4 Identification of USP7 substrates in a time-course experiment. a** Schematic of the USP7 inhibitor time-course experiment. HCT116 cells were treated with DMSO or 10 μM of FT671 and harvested in SDC buffer at the indicated time points. After tryptic digestion of 2 mg of total protein per sample, both the ubiquitinome and the proteome were acquired in single-shot DIA mode and the resulting raw files processed using DIA-NN. **b** Volcano plots of the ubiquitinome (15 min) and the proteome (6 h) after FT671 treatment are shown on the left. Significantly-regulated proteins and K-GG peptides (LIMMA[52], 5% FDR) are colored (Log2 fold changes $-1 < x < 1$ in blue and $x < -1$ or $x > 1$ in red). The Venn diagram shows significantly and >2-fold ubiquitinated proteins at 15 min (red), and ubiquitinated proteins that were significantly downregulated by more than 20% over the 6 h time course (blue). The overlapping proteins were significantly enriched for categories such as zinc finger and ubiquitin-protein ligase activity. BH FDR = Benjamini–Hochberg false discovery rate. **c** Heat map of proteins that were significantly downregulated by more than 20% upon FT671 treatment (or more than 50%, in magenta) and that showed significant, and more than twofold induction of at least one ubiquitination site at 15 min. Ubiquitin peptide profiles were averaged and both data were matched based on gene level. Topors was included in the heatmap - its protein intensities were below detection limit (missing values, in gray) because it was rapidly degraded upon FT671 treatment. **d** Profiles of Log2 fold changes (FT671/DMSO) at different time points for different cell cycle regulators upon USP7 inhibition. Plotted are mean ± SEM. $n = 4$ biological replicates. **e** After transfection with different siRNAs (Control, USP7_1, USP7_2) for 48 h, HCT116 cells were treated with DMSO/10 μM FT671 for 5 min. Shown are Log2 fold changes (FT671/DMSO) of all quantified K-GG remnants for siRNAs targeting USP7 (y-axis on both plots) or a control siRNA (x-axes). Selected targets found to be regulated in the time course experiment (Fig. 4a–d) are colored. **f** Significantly upregulated ubiquitinated peptides at 15 min of FT671 treatment were mapped onto a BioGrid[37] network for reported USP7 interacting proteins (filtered for four evidences). The proteins were colored according to the peaking time of the ubiquitin signal (averaged profiles for significantly upregulated peptides). Source data are provided as a Source Data file.

twofold induced, with the strongest up-regulation seen for 22 K-GG peptides originating from USP7 itself, pointing at auto-regulation. Orthogonal validation by enriching ubiquitinated proteins with ubiquitin-binding domains, followed by either Western blotting or MS analysis, independently confirmed these results (Supplementary Fig. 7). We further detected 151 ubiquitinated peptides mapping to 33 of those proteins that were ubiquitinated and degraded at later time points (15 min–6 h, Fig. 4), again suggesting that these proteins are bonafide USP7 targets (Supplementary Fig. 6b and Supplementary Data 10).

**Profiling of multiple USP7 inhibitors reveals a high overlap of regulated ubiquitination sites.** Structurally distinct compounds targeting USP7 likely have different off-target effects. To determine such effects and to strictly define USP7 targets, we next analyzed the ubiquitinome dynamics upon treatment with four different USP7 inhibitors, namely FT671, FT827, GNE-6640 and GNE-6776[30,33]. Although PCA clearly separated the different compounds from each other, we found an overall good correlation of the significantly-regulated ubiquitinated peptides, reflecting USP7 inhibition as the shared mode-of-action of the compounds. As expected, the treatments with the structurally similar compounds GNE-6640 and GNE-6776 overall correlated best with each other. Furthermore, GNE-6776, but not GNE-6640 effects, correlated well with both FT671 and FT827, indicating a greater difference in target selectivity (Fig. 5a–c). Interestingly, several ubiquitination sites on USP7 were strongly induced by both FT671 and FT827 and much less by GNE-6640 and GNE-6776, probably reflecting a higher potency[30,33] of the FT compounds (Fig. 5d). Finally, to get a high-confidence list of primary USP7 substrates targeted for proteasomal degradation, we determined the overlap of degraded ubi-proteins in our time course experiment and ubiquitinated proteins with sites significantly and more than twofold increased by three out of the four tested inhibitors. This revealed an overlap of 70%, further strengthening the evidence that these are direct enzyme targets directed for degradation (Fig. 5e and Supplementary Data 11).

## Discussion

MS-based ubiquitinomics has seen enormous improvements over the past years. Initially identifying just a handful of ubiquitination sites, more recently the quantification of thousands of sites to decipher ubiquitin signaling became commonplace[6,26,38–40]. Until now, exclusively data-dependent acquisition mass spectrometry in combination with label-free, SILAC or TMT

quantification has been explored to address challenging biological questions. Here we introduced an improved workflow featuring an SDC-based lysis protocol, combined with DIA-MS and deep neural network-based data processing using DIA-NN. We not only identify very high numbers of ubiquitinated peptides, but also quantify a large fraction of those with high precision. Another strength of our method is the very high data completeness, a prerequisite for robust statistical analyses, in particular in medium- and large-scale experiments. Similar to this study, in a simultaneously preprinted work, Mann and colleagues recently reported the advantages that DIA-MS offers over label-free DDA in ubiquitinomics[16]. However, despite using shorter chromatographic gradients, our improved lysis protocol along with our deep neural network-based processing software (DIA-NN), which we specifically optimized for ubiquitinomics, allowed us to double the numbers of quantified K-GG peptides.

DIA-based experiments typically employ spectral libraries, which are built by either high pH reversed-phase or gas-phase fractionation of peptides[23,41,42]. Library-free searches instead use in silico generated spectral libraries, which are particularly advantageous in case of PTM analyses, where experimental library generation can be very laborious and require large protein amounts[23]. We demonstrate that library-free DIA can perform comparably well to a DIA analysis workflow based on experimentally-generated spectral libraries, and rigorously validate its identification confidence as well as its quantification accuracy. Without the need for a dedicated spectral library, our workflow is ideally suited for applications with minimal sample amounts, such as tumor biopsies, primary cells or laser micro-dissected tissue samples[43]. Nevertheless, for screening-type applications, we recommend generating a spectral library, which speeds up data processing considerably without compromising on analytical depth.

The gain in sensitivity, throughput, and quantitative precision achieved by our workflow allowed us to introduce an advanced strategy for probing DUB substrates: upon chemical enzyme inhibition, we track the dynamic changes in both the proteome and the ubiquitinome at high temporal resolution. By doing so, early-induced ubiquitination events on a target protein can be connected to its degradation at later time points, allowing the pinpointing of putative substrates with high confidence. Nevertheless, to confirm direct DUB targets, orthogonal assays such as in vitro deubiquitination assays or proximity-labeling experiments are required. Besides available inhibitors of the UPS, gene expression knockdown technologies such as CRISPRi[44], in combination with our DIA-MS ubiquitinomics workflow, could also

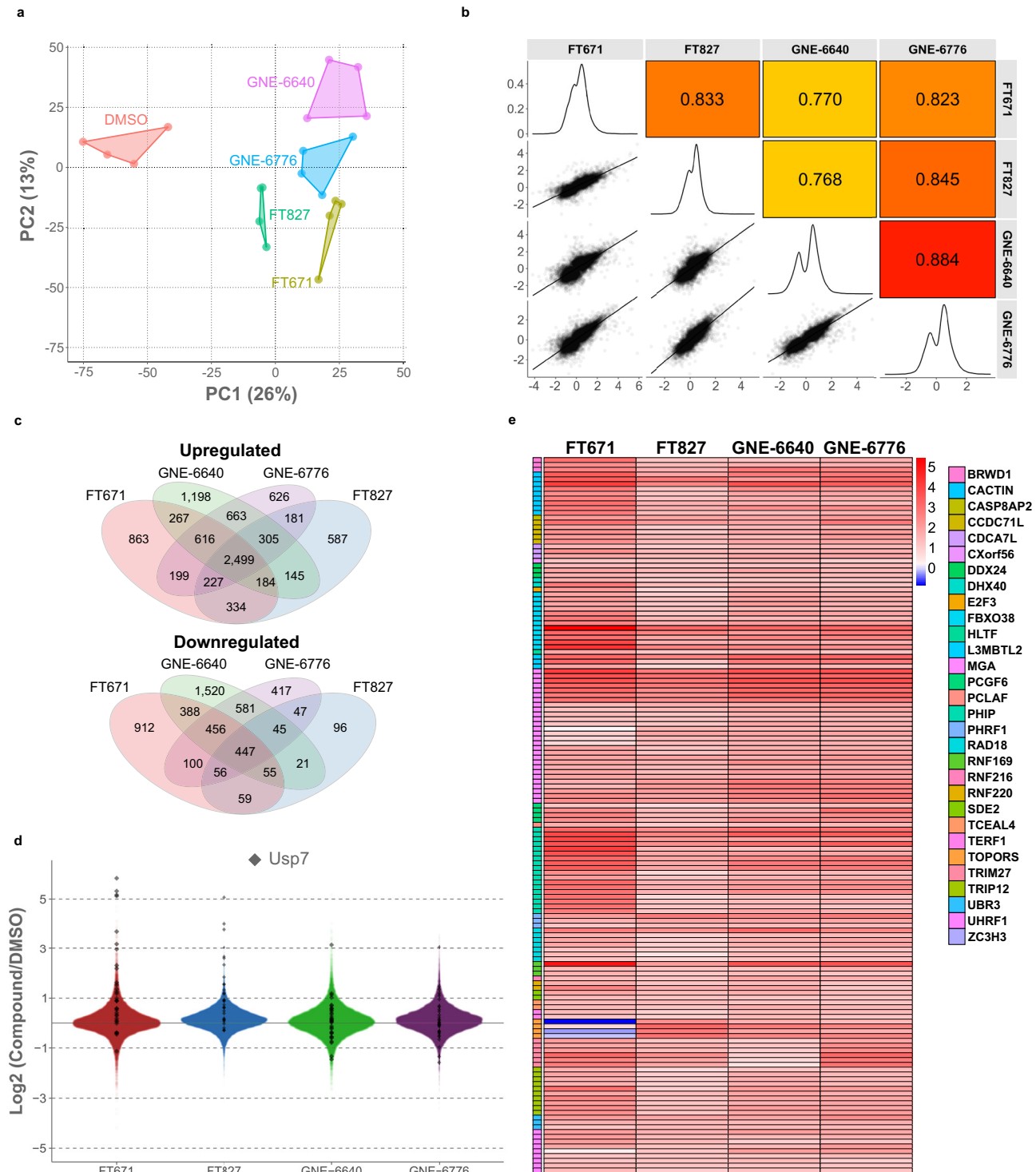

**Fig. 5 Profiling of multiple USP7-targeting compounds by DIA ubiquitinomics. a** HCT116 cells were treated with four structurally distinct USP7 inhibitors for 1 h. Shown is a principal component analysis (PCA) of the ubiquitinomics data. **b** Log2 fold changes (lower left part), density distributions (middle part), and Pearson correlations of significantly regulated K-GG peptide intensities with four different compounds targeting USP7. **c** Venn diagrams of significantly up- and downregulated K-GG peptides with the different compounds. **d** Dot plot showing Log2 fold changes (USP7-targeting compound vs DMSO) of K-GG peptides for FT671, FT827, GNE-6640, and GNE-6776. K-GG peptides mapping to USP7 are highlighted. **e** Fold changes of ubiquitinated peptides (FT671/DMSO, Log2-transformed) for four structurally distinct USP7 inhibitors. Shown are ubiquitin peptide profiles for those proteins that harbor significantly and >2-fold increased ubiquitination sites with three out of four USP7 inhibitors and that are ubiquitinated and degraded upon FT671 treatment (see Fig. 4). Source data are provided as a Source Data file.

be leveraged for future systematic DUB/E3 ligase substrate screens.

We applied our method to characterize USP7 targets in an unbiased manner. Besides confirming known substrates, such as Mdm2, we also mapped actual modification sites on these proteins. On top of that, we uncovered a set of transcriptional regulators and E3 ligases as putative substrates directed for proteasomal degradation. Interestingly, using a combination of different inhibitors and MS-based proteomics to detect down-regulated proteins (without concomitant ubiquitinome profiling), a recent report identified similar proteins as bona fide USP7 targets, namely a number of DNA repair enzymes and E3 ligases[45]. While protein abundance changes can in theory be used as a readout for mapping DUB enzyme targets, a major limitation of such an approach is that direct targets cannot be distinguished from secondary regulations, and regular proteomics does not allow identifying ubiquitination sites. Our method overcomes this hurdle by simultaneously recording protein ubiquitinition and protein abundances.

We observed an ultra-fast response after USP7 inhibition, with about 1,000 upregulated ubiquitinated peptides mapping to hundreds of proteins, even at the earliest time point tested (2 min, Supplementary Data 10). Surprisingly, only a very small subset of these targets was actually degraded at later time points, and this raises several questions for follow-up studies. For example, what are the ubiquitin chain types that USP7 acts upon or what ubiquitination threshold must be reached before a protein is targeted for degradation by the proteasome. Another open question is whether DUBs other than USP7 have a similar large target spectrum.

In conclusion, our optimized ubiquitinomics workflow accurately and consistently quantifies tens of thousands of ubiquitinated peptides at high throughput and enables precise time-resolved analysis of in vivo ubiquitination dynamics. We mapped hundreds of ubiquitination sites on both known and not reported USP7 substrates and determined their impact on protein abundance regulation.

Besides direct enzyme-substrate relationship investigation, our method will be useful to define modes-of-action of drugs directed against different components of the UPS. Furthermore, it will aid the better understanding of ubiquitin signaling in processes such as DNA repair or protein/organellular turnover, which can lead to cancer formation or neurodegeneration, if dysregulated.

## Methods

**Reagents.** MG-132 (474787), Bortezomib (5043140001), 2-chloroacetamide (CAA, 22790), tris(2-carboxyethyl)phosphine hydrochloride (TCEP, C4706), sodium deoxycholate (SDC, 30970), dimethyl pimelimidate dihydrochloride (DMP, D8399), ethanolamine (411000), Na$_2$HPO$_4$ (S9763), 3-(N-morpholino)propane-sulfonic acid (MOPS, M5162), sodium azide (S2002), N-ethylmaleimide (E3876), PR-619 (SML0430) and sodium tetraborate decahydrate (S9640) were from Millipore Sigma. Ethylenediaminetetraacetic acid (EDTA, 108418), Tris(hydroxymethyl)-aminomethan (Tris, 108382), trifluoroacetic acid (TFA, 108178), ammonium hydroxide (NH$_4$OH, 533003), sodium chloride (106404), ethyl acetate (103649), acetonitrile (ACN, 100030) from Merck. RNAiMAX from Thermo Scientific (13778075). Formic acid (FA, 56302) from Fluka. BSA (01400) from Biomol. FT671 (5-((1-(4,4-Difluoro-3-(3-fluoro-1H-pyrazol-1-yl)butanoyl)-4-hydroxypiperidin-4-yl)methyl)-1-(4-fluorophenyl)-1,5-dihydro-4H-pyrazolo[3,4-d]pyrimidin-4-one, AOB37855, Purity =98% by HPLC), GNE-6640 (4-[2-Amino-4-ethyl-5-(1H-indazol-5-yl)-3-pyridyl]phenol, AOB37854, Purity =98% by HPLC) and GNE-6776 (6′-Amino-4′-ethyl-5′-(4-hydroxyphenyl)-N-methyl-[3,3′-bipyridine]-6-carboxamide, AOB37852, Purity = 98% by HPLC) from AOBIOUS, FT827 (Ethenesulfonamide, N-[4′-[[4-[(1,4-dihydro-1-methyl-4-oxo-5H-pyrazolo[3,4-d]pyrimidin-5- yl)methyl]-4-hydroxy-1-piperidinyl]carbonyl][1,1′-biphenyl]-2-yl]-, HY-111350, Purity= 98.48% by LC/MS) from MedChemExpress and Trypsin/LysC mix (V5071 or V5072) from Promega. PTMScan® Ubiquitin Remnant Motif (K-ε-GG) Kit (#5562) from Cell Signaling Technology. The mix of hundred synthetic K-GG peptides was obtained from JPT peptide technologies (SpikeMix™ PTM-Kit 47 - Lys(GG)) and was resuspended in 100 μl of 0.1% formic acid by sonication. The tryptic yeast digest from Promega (V7461) and the E. coli digest from Waters (186003196). Both were resuspended in 100 μl of 0.1% formic acid.

**Cell culture, drug treatments, transfection, and cell lysis.** HCT116, MM.1S, and Jurkat 6.1 were from ATCC. HCT116 were cultured in DMEM, 10% FCS, 4 mM L-glutamine and 1 mM sodium pyruvate. Jurkat 6.1 and MM.1S were grown in RPMI, 10% FCS, 2 mM L-Glutamine, and 1 mM Sodium pyruvate. MG-132, FT671, FT827, GNE-6640, and GNE-6776 were dissolved in DMSO to prepare a 10 mM stock solution. Cells were treated with either DMSO or the indicated compounds, washed twice with ice-cold PBS, and harvested using freshly prepared SDC buffer (room temperature) or urea buffer (chilled to 4 °C). The SDC buffer contained 1% SDC, 10 mM TCEP, 40 mM CAA, 75 mM Tris-HCl at pH = 8.5 and its pH was adjusted to 8.5 with 1 N NaOH. The urea buffer contained 8 M urea, 1 mM EDTA, 1 mM CAA, 150 mM NaCl, 50 mM Tris-HCl pH 8.0, 2 μg/ml aprotinin, 10 μg/ml leupeptin, 50 μM PR-619 and 1 mM PMSF, which was added immediately before use. The SDC lysates were heated to 95 °C for 10 min while shaking at 750 rpm in a Thermomixer (Eppendorf) and then sonicated for 10 min (10 × 30 s on/off cycles) using a Bioruptor® Pico sonication device (Diagenode) (for volumes <2 ml) or a ultrasonic probe device (Bandelin Sonopuls, 1 min, energy output ~40%, for volumes >2 ml). For the SDC/urea comparison in Fig. 1, the urea lysates were processed as described previously[18]. Briefly, the cell pellet was homogenized by pipetting up and down and the lysate cleared by centrifugation at 20,000 × g for 10 min at 4 °C. The supernatant was then transferred to a new tube.

For siRNA-mediated knockdown of USP7, HCT116 cells were transfected with either a control siRNA (QIAGEN, 1022076) or two different siRNAs targeting USP7 (siUSP7_1= QIAGEN FlexiTube, 1027417, siUSP7_2= Sigma Aldrich, EHU131171) with RNAiMAX (Thermo Scientific), following the vendor's protocol. The cells were transfected twice (at day 1 (0 h) and day 2 (24 h)) and drug treatment (DMSO or 5 μM FT671 for 5 min) followed by cell lysis performed at 48 h post-transfection.

**Ubiquitin domain pulldown experiments.** HCT116 cells were treated with 10 μM MG-132, alone or in combination with 10 μM FT671 for 15 min. Cells were harvested in ice-cold lysis buffer (50 mM Tris-HCl pH 7.5, 0.15 M NaCl, 1 mM EDTA, 1% NP-40, 10% glycerol, freshly supplemented with 50 μM PR-619 and 10 mM NEM (N-ethylmaleimide) and spun at 14,000 g for 10 min. The supernatant was transferred to a fresh tube and the protein concentration determined using the BCA assay (Thermo Scientific, 23225). 20 μl of TUBE2 agarose matrix (LifeSensors, UM402) were added to 2 mg of total lysate, followed by incubation at 4 °C for 1 h on a rotor wheel. The beads were collected by centrifugation and washed three times each with 1 ml of TBS (20 mM Tris-HCl pH 8.0, 0.15 M NaCl)-0.1% Tween-20. The proteins were eluted using a 5× SDS-sample buffer (200 mM Tris pH 6.8, 5% (v/v) ß-Mercaptoethanol (freshly added), 5% (w/v) SDS, 50% Glycerol, 0.25% (w/v) Bromphenolblau), separated on SDS-PAGE followed by Western blotting. In short, proteins were transferred to a nitrocellulose membrane and the membrane was blocked in 5% milk/TBS-0.1% Triton for 1 h at room temperature. After washing for three times with TBS-0.1% Triton, the membrane was incubated overnight with a primary USP7 antibody (Cell signaling technology, 4833, 1:1000 in 2% BSA-TBS-0.1% Triton) at 4 °C. The following day, the membrane was washed, incubated with a secondary HRP-linked (Rockland (611–1302), 1:10,000 in 2% BSA-TBS-0.1% Triton) anti-rabbit antibody and USP7 detected by chemiluminescence (ChemiDoc, Biorad). For MS analysis, proteins were digested on-bead. After washing with 3 × 1 ml TBS-0.1% Tween-20 and 1 × 1 ml TBS, 100 μl of SDC buffer were added to the beads, followed by incubation at 95 °C for 10 min. After cooling to RT, 0.5 μg of trypsin was added to each sample, followed by overnight incubation at 37 °C in a thermomixer. The following day, 100 μl of 99% isopropanol/1% TFA were added and the peptides were desalted using SDB-RPS STAGE tips. STAGE tips were activated using 100 μl of 99% isopropanol/1% TFA, then washed with 75 μl of 80% ACN/5% NH$_4$OH, followed by two washes with 99% isopropanol/1% TFA (75 μl each). The samples were then loaded onto the tips and washed twice with 99% isopropanol/1% TFA (200 μl each) and additional two times with 0.2% TFA (200 μl each). Finally, the peptides were eluted with 75 μl of 80% ACN/5% NH$_4$OH, speed-vac dried, and resupended in 20 μl of 0.1% formic acid. After peptide concentration estimation on the nanodrop (Thermo Scientific), the concentration was set to 0.4 μg/μl and 2 μl were injected into the mass spectrometer (800 ng).

**Crosslinking of K-GG antibody.** Crosslinking was performed as described previously[18]. In brief, antibody-bound beads were washed with 3 × 1 ml of 100 mM sodium tetraborate (pH 9.0) and then crosslinked by incubating with 1 ml of 20 mM DMP/100 mM sodium borate (pH = 9.0) for 30 min at room temperature. The crosslinking buffer was removed and the reaction stopped by washing with 2 × 1 ml of 200 mM ethanolamine (pH = 9.0), followed by incubation with 1 ml of 200 mM ethanolamine (pH = 9.0), with end-over-end rotation at 4 °C for 2 h. Finally, the beads were washed with 3 × 1 ml of IP buffer and either directly used or conserved for up to 2 weeks in 1× phosphate-buffered saline (PBS)/0.02% sodium azide.

**K-GG peptide enrichment and LC-MS/MS sample preparation**. Protein concentrations were determined using the BCA assay (for experiments in Fig. 1a) or the 660 nm assay (all other experiments, Thermo Fisher Scientific). 2 mg of urea lysate for each replicate was reduced with 5 mM DTT for 45 min and then alkylated with 10 mM CAA for 30 min in the dark before digestion. Proteins were digested with trypsin/Lys-C mix overnight at 37 °C (SDC lysates) at room temperature (urea lysates) with a 1:50 enzyme to protein ratio. For urea samples, the digestion was stopped by adding TFA to a final amount of 0.1% (v/v) and peptides were desalted using C18 cartridges (Sep-Pak tC18, WAT036790) as follows: (a) conditioning with 5 ml of ACN; (b) conditioning with 5 ml of 50% ACN/0.1% FA; (c) equilibration with 4 × 5 ml of 0.1% TFA; (d) loading of the sample, (e) washing with 4 × 5 ml 0.1% TFA, (f) elution with 2 × 3 ml 50% ACN/0.1% FA. For SDC-lysed samples, the digestion was stopped by adding two volumes of 99% ethyl-acetate/1% TFA, followed by sonication for 1 min using an ultrasonic probe device (energy output ~40%). The peptides were desalted using 30 mg (for < 1 mg of input, 8B-S029-TAK) or 100 mg (for up to 2 mg of input, 8B-S029-EBJ) Strata-X-C cartridges (Phenomenex) as follows: (a) conditioning with 1 ml/3 ml (for 30 mg and 100 mg cartridges, respectively) of isopropanol; (b) conditioning with 1 ml/3 ml of 80% ACN/5% NH$_4$OH; (c) equilibration with 1 ml/3 ml of 99% ethyl-acetate/1% TFA; d) loading of the sample; e) washing with 2 × 1 ml/3 ml of 99% ethylacetate/1% TFA; f) washing with 1 ml/3 ml of 0.2% TFA; g) elution with 2 × 1 ml/3 ml of 80% ACN/5% NH$_4$OH. The eluates were snap-frozen in liquid nitrogen and lyophilized overnight. K-GG peptide enrichment was performed as described with some minor modifications[18]. Briefly, peptides were resuspended in 1 ml of cold immunoprecipitation (IP) buffer (50 mM MOPS pH 7.2, 10 mM Na$_2$HPO$_4$, 50 mM NaCl) and incubated with 40 µl of a 25% slurry of cross-linked K-GG antibody-bead conjugate (corresponding to 10 µl beads/IP) for 2 h at 4 °C with end-over-end rotation. Beads were then washed twice with 1 ml of IP buffer and an additional time with cold Milli-Q® water. After removing all the supernatant, the beads were incubated with 200 µl of 0.15 % TFA at room temperature while shaking at 1,400 rpm. After briefly spinning, the supernatant was recovered and desalted using in-house prepared, 200 µl two plug StageTips[46] with SDB-RPS (3 M EMPORE™, 2241; for SDC) or C18 (3 M EMPORE™, 2215; for urea). SDB-RPS StageTips were conditioned with 60 µl isopropanol, 60 µl 80% ACN/5% NH$_4$OH, and 100 µl 0.2% TFA. The K-GG enrichment eluate (0.15% TFA) was directly loaded onto the tips followed by two washing steps of 200 µl 0.2% TFA each. Peptides were eluted with 80% ACN/5% NH$_4$OH. For C$_{18}$, StageTips were equilibrated with 60 µl ACN and 2 × 60 µl 0.1% FA before sample loading. StageTips were washed with 2 × 60 µl or 0.2% TFA and peptides eluted with 60 µl of 50% ACN/0.1% FA. Peptides were Speedvac dried and then resuspended in 10 µl of 0.1% FA, of which 4 µl were injected into the mass spectrometer. For total proteome measurements, a 50 µl aliquot of desalted peptide eluate was transferred to a 0.5 ml tube, Speedvac dried and resuspended in 15 µl of 0.1% FA. The peptide concentration was estimated using a Nanodrop™ device (Thermo Fisher Scientific) and adjusted to 0.4 µg/µl with 0.1% FA, of which 2 µl (800 ng) were injected into the mass spectrometer.

**High-pH reversed-phase fractionation**. HCT116 cells were treated with MG-132 (10 µM) or FT671 (10 µM) for 6 h before lysis with SDC buffer (see "Cell culture, drug treatments and cell lysis" for details). 40 mg of each lysate were combined before overnight digestion (37 °C) with Trypsin/LysC in a 1:50 enzyme to protein ratio. The resulting peptides were desalted using Strata-X-C cartridges (Phenomenex) as described in "K-GG peptide enrichment and LC-MS/MS sample preparation". The lyophilized peptides were resuspended in 0.1% FA and fractionated using a Zorbax 300SB-C18 column (Agilent Technologies) on an ÄKTA HPLC system (GE Healthcare). Fractionation was performed with a flow rate of 3 ml/min and with a constant flow of 10% 25 mM ammonium bicarbonate, pH 10. Peptides were separated using a linear gradient of ACN from 5% to 35% over 45 min, followed by a 5-min increase to 60% ACN and ramping to 70% over 3 min. Fractions were collected at 60-s intervals in a 48-well plate to a total of 36 fractions and then pooled to obtain 12 fractions (A1-C1-E1, A2-C2-E2 etc.). All fractions were acidified by adding FA to a final amount of 0.1% and then lyophilized. Peptides were subsequently resuspended in 1000 µl 0.1% TFA and desalted using Strata-X-C cartridges (Phenomenex) as described. Lyophilized peptides of each fraction were resuspended in 1 ml of IP buffer and enriched for K-GG peptides. Each fraction was injected twice into the mass spectrometer (36 measurements in total) and the data were acquired with a 75 min LC-MS method (DIA). The data were processed in DIA-NN, using the library-free setup.

**LC-MS/MS measurements**. Peptides were loaded on 40 cm reversed-phase columns (75 µm inner diameter, packed in-house with ReproSil-Pur C18-AQ 1.9 µm resin [ReproSil-Pur®, Dr. Maisch GmbH]). The column temperature was maintained at 60 °C using a column oven. An EASY-nLC 1200 system (ThermoFisher) was directly coupled online with the mass spectrometer (Q Exactive HF-X, ThermoFisher) via a nano-electrospray source, and peptides were separated with a binary buffer system of buffer A (0.1% formic acid (FA) plus 5% DMSO) and buffer B (80% acetonitrile plus 0.1% FA plus 5% DMSO), at a flow rate of 250 nl/min (for 75 min and 125 min gradients). For the 60 min method, a flow rate of 300 nl/min was used. For the 30 min and 45 min gradients, a flow rate of 350 nl/min was used and for the 15 min DIA method, a 15 cm column and a flow rate of

500 nl/min were used. The mass spectrometer was operated in positive polarity mode with a capillary temperature of 275 °C.

The DDA method consisted of a MS1 scan ($m/z = 300$–1650, $R = 60,000$, maximum injection time= 20 ms, spectrum data type = profile) followed by TopN MS/MS scans ($N = 15$). These were acquired at $R = 15,000$, AGC target = 1e5, maximum injection time= 28 ms, isolation window 1.4 Th, NCE = 27 and a scan range of 200–2000 $m/z$.

The DIA methods consisted of a MS1 scan ($m/z = 300$-1,650) with an AGC target of $3 \times 10^6$ and a maximum injection time of 60 ms ($R = 120,000$). DIA scans were acquired at $R = 30,000$, with an AGC target of $3 \times 10^6$, "auto" for injection time and a default charge state of 4. The spectra were recorded in profile mode and the stepped collision energy was 10% at 25%. The number of DIA segments was set to achieve an average of 4–5 data points per peak (Supplementary Data 4).

**Scoring of ubiquitination sites implemented in DIA-NN**. A set of scores that reflect MS2-level evidence for detection of the modified peptide were implemented in DIA-NN. In DIA proteomics, peptides are matched not just to a single spectrum, but to a series of consecutive spectra, wherein each peptide fragment ion gives rise to an elution profile. In the absence of interfering signals, originating from co-fragmenting peptides that share fragments with the same masses, the elution profiles of fragments of a peptide correlate. Detecting these correlations is the primary method that enables matching peptides to the data in DIA proteomics[24,47,48]. In DIA-NN, this concept is explored to boost the confidence of K-GG peptide identification. When assessing the candidate elution peaks, which could potentially originate from a particular peptide, DIA-NN always designates one of its fragments as the "best", based on which fragment has an elution profile that best correlates with elution profiles of other fragments. The elution profile of the "best" fragment is then considered representative of the true elution profile of the peptide itself[24]. The "quality" of signals corresponding to all other fragments as well as to the extracted elution profile of the precursor itself (MS1 level) is then assessed based on how well their elution profiles correlate with the elution profile of the "best" fragment, with high correlation being indicative that the fragments likely originate from the same precursor. DIA-NN leverages this principle for confident K-GG detection. A number of scores are calculated, using sums of such correlations, for all the fragments that contain/do not contain the modified residues/other potential modification sites (in different combinations of these conditions), and used to determine the confidence of modified peptide identification (Supplementary Fig. 8).

**Raw data processing**. MS raw files acquired with DDA mode were analyzed using MaxQuant[20], version 1.6.17.0 (maxquant.org), whereas files acquired in DIA mode were processed using DIA-NN[24], version 1.8 (https://github.com/vdemichev/DIA-NN). Reviewed UniProt entries (human, SwissProt 11-2020 [9606], 42,395 entries; yeast [559292], SwissProt 11-2020, 6,750 entries and *E. coli* [83333], SwissProt 01-2021, 4,532 entries) were used as protein sequence database for both MaxQuant and DIA-NN searches. For MaxQuant, the standard settings were used (enzyme = trypsin/P, two missed cleavages, carbamidomethylation of cysteines as fixed modification and protein N-terminus acetylation and oxidation of methionine as variable modifications) and K-GG was added as a variable modification. "Match between runs" (MBR) was enabled where indicated. For DIA-NN, one missed cleavage and a maximum of two variable modifications per peptide were allowed (acetylation of protein N-termini and oxidation of methionine). Carbamidomethylation of cysteines was set as fixed modification and K-GG was added in case of ubiquitinomics. Unless specified, all ubiquitinomics data analyses were carried out using library-free analysis mode in DIA-NN. All proteome data analyses were performed in library-free mode. For details on setup of DIA-NN searches, see Supplementary Fig. 9. For library-free searches, "deep learning-based spectra and RTs prediction" was enabled. MBR was enabled. This instructs DIA-NN to follow the two-step workflow as described previously[24]. Briefly, DIA-NN first generates a spectral library from DIA data using all identifications in the specified raw files. The library generation is performed using both global (experiment-wide) and run specific precursor FDR filters of 1%. This library is then used as a spectral library in a second search. The quantification strategy was set to "Robust LC (high precision)". Spectronaut (v. 14.10.201222.47784) was run with default settings, except PTM localization was enabled and set to 0, while protein q-value filtering was disabled.

**Bioinformatic data analysis**. Modification-specific peptides of MaxQuant outputs (DDA) were filtered to exclude reverse hits and contaminants. Counts of K-GG and non-K-GG peptides, missing values, and CVs were computed using the Perseus software[49]. Specifically, the modification-specific peptide output table from MaxQuant was filtered for unique peptide sequences for both K-GG and non-K-GG peptides.

For DIA-NN ubiquitinomics outputs, precursors were aggregated to peptides by averaging normalized precursor intensities. Coefficients of variation were computed on the K-GG peptide raw intensities and the plots were created with R (version 3.6.1). The GO term enrichment analysis was done with Perseus[49].

DIA-NN outputs for Figs. 4 and 5 and associated supplementary figures were further processed with R. K-GG precursor intensities were aggregated to peptides using the MaxLFQ[50] algorithm, as implemented in the DIA-NN R package (https://github.com/vdemichev/DIA-NN-rpackage/). K-GG peptide to site

mapping was done using reviewed entries of the human UniProt database (SwissProt, release 11-2020, 42,395 entries). The peptide intensities were normalized by median sample scaling before differential expression analysis.

The proteome raw data were processed with the library-free search modey in DIA-NN. Instead of using the protein groups created by DIA-NN, protein inference was performed in-house following the logic of the ID Picker[51] algorithm.

Significance testing of log2-transformed intensities was performed with LIMMA[52] on all peptides/proteins present in more than 50% of samples; inhibitor treatments were compared to their corresponding DMSO controls at each time point. Multiple testing-corrected q-values <0.05 were considered as significant. For comparing proteome and ubiquitinome data, protein groups of each dataset were disaggregated into individual UniProt identifiers and then mapped between the data sets. For individual plots, the protein groups of the ubiquitinome data sets along with K-GG site positions were used. Hierarchical clustering was performed using Ward's method on Euclidean distances, separately for proteomes and ubiquitinomes. The data was visualized in Spotfire (version 7.12.0).

Cytoscape (version 3.7.2, cytoscape.org) was used for mapping the combined interactomics data from BioGrid[37] to the ubiquitinomics data from this study. BioGrid interaction data (version 3.5.186) was filtered for human proteins with physical interactions with USP7 in at least four studies. Ubiquitination sites were mapped onto USP7 and its interactors, if they showed significant upregulation at 15 min. Proteins without significant ubiquitination regulation are not shown.

**Reporting summary**. Further information on research design is available in the Nature Research Reporting Summary linked to this article.

## Data availability

The mass spectrometry proteomics raw data and the corresponding processing reports generated in this study have been deposited to the ProteomeXchange Consortium (http://proteomecentral.proteomexchange.org) via the PRIDE[53] partner repository with the dataset identifier PXD023889. Previously published data used in various benchmark experiments are available under the accession codes PXD019854 and PXD006201. Source data are provided with this paper.

## Code availability

DIA-NN is freely available for download at https://github.com/vdemichev/diann.

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

## Acknowledgements

We thank all members of Evotec München GmbH and especially S. Blencke, A. Kirgis, T. Lanzinner, J. Linnemann, S. Marx, J. Modrow, R. Jörg, R. Putzhammer and D. Turnblad-Phillips for technical assistance. M. Segura-Lepe, D. Bader, C. Kluger and B. Kracher for help in developing the DIA data analysis pipeline. J. Drechsel for help with drawing of chemical structures. A. Beling, T. Gaitanos and M.G. Steger for proofreading the manuscript. M. Mann, F. Hansen and Ö. Karayel for sharing raw data and for exchange of ideas.

## Author contributions

M.S. conceived the study. H.D., S.M., M.S. and V.D. designed the experiments. M.B., P.I., M.S. and V.D. analyzed the data. V.D. developed the DIA-NN software. U.O. and M.S developed the DIA methods. M.S. coordinated the studies. All authors interpreted the data. M.S., H.D., V.D. and M.R. wrote the manuscript and S.M. proofread and edited the manuscript. All authors read and approved the final manuscript.

## Competing interests

V.D. and M.R. declare no competing interests. M.S., M.B., S.M., U.O. and H.D., are employees of Evotec München GmbH (Martinsried, Germany). P.I. is an employee of Roche. H.D. has share options of Evotec AG.
