## [Peer Review File · Nature Communications]

REVIEWER COMMENTS

Reviewer #1 (Remarks to the Author):

The authors of this manuscript, “Time-resolved in vivo ubiquitinome profiling by DIA-MS reveals Usp7 targets on a proteome-wide scale”, present a novel mass spectrometry-based approach for studying ubiquitinated proteins, and apply their approach to study the effects of several inhibitors of the deubiquitinase Usp7 on protein degradation. The authors sufficiently demonstrate the improvements made by their proposed ubiquitinome protocol, although unfortunately much of the novelty of using SDC and DIA has been recently “scooped” by the Hansen et al 2021 Nat Comms paper published in this journal. That said, there is quite a bit of informative, but downplayed, data on the implications of using DIA versus DDA that might be expanded to regain some novelty in that regard. Overall, the authors seem quite knowledgeable of the mass spectrometry-based ubiquitination field and have presented interesting datasets. I imagine the manuscript’s conclusions will be of interest to mass spectrometrists performing ubiquitin PTM work, and perhaps also assist drug discovery efforts geared towards degraders such as the ubiquitin-proteasome pathway.

I recommend publication of the manuscript with the following major and minor revisions to support the interpretation and the presentation of the manuscript’s main points.

Major issues:

(1) Although the authors do mention the Hansen et al preprint in the Discussion, that work has now been published in this journal (Hansen et al 2021, Nat Comms). As the Hansen paper also used SDC and DIA for ubiquitination, the authors should adjust their language appropriately (e.g. use of “novel” is already a bit dated, as Hansen et al did it “first”). However, there is much room to compare/contrast the salient points of this work versus Hansen et al. As one example, Hansen et al relies on Spectronaut for DIA analysis, which is a closed-source and proprietary software, whereas the authors here are using a free and open-source software, DIA-NN. A few more detailed points below:

(1a) Although the authors have done well to benchmark DIA-NN for ubiquitin modifications, there is no comparison of DIA-NN versus other software. I would like to see a comparison of DIA-NN versus Spectronaut (for the sake of comparing to Hansen et al) or, if a Spectronaut license is not possible, some other DIA software like IPF (Rosenberger et al 2017, Nat Biotech) or Thesaurus (Searle et al 2019, Nat Methods).

(1b) The authors claim novelty with a library-free DIA performing comparably to DIA analysis based on experimentally-generated spectral libraries, e.g. in the Methods, “Briefly, DIA-NN first generates a spectral library from DIA data using all identifications in specified raw files.” I would like a bit more elaboration on this point, as to my knowledge none of the other predicted spectral library algorithms (MS2PIP, ProSight) can generate modified peptide spectra. Does DIA-NN work for other modifications as well? Does it predict retention times, like other predicted spectral library tools? What about multiply-modified peptides?

(2) The authors use an unusual type of plot in Figures 4B and S3. These types of figures are usually shown as volcano plots, but here the author is using “standard error” instead of “p-value” on the vertical axis. I request the authors provide volcano plots for these figures, at least as supplemental figures, because these plots are so unusual for proteomics data.

Minor points:

- Pg 3 paragraph 2, “For example, while K11- and K48-linked chains can tag proteins for proteasomal degradation, ubiquitin conjugates on K63 tend to modulate protein-protein interactions”. It’s unclear if this refers to some specific protein with ubiquitinated K11, K48, and K63 residues, or if this means that the lysine can have 11-, 48-, or 63-ubiquitin moiety-long chains on a lysine.

- Pg 3, paragraph 4, “By introducing an SDC-based lysis protocol [...]” the abbreviation “SDC” should be defined first before using the abbreviation.

- Pg 3, paragraph 4, “[...] as well as significantly improve quantitative accuracy.” The authors show quantitative precision/reproducibility, but only show better accuracy in figure S7 which is actually a very interesting figure that isn’t mentioned at all in the text! Authors should use some text to describe that DIA improves quantitative accuracy per Fig S7.

- Fig 1B - on average, how many lysine residues are ubiquitin-modified per protein? One ubiquitinated lysine per protein? Two? Six?

- Pg 7, paragraph 2, “Although DIA-NN uses a rigorous approach to determine the false discovery rate (FDR) [...]” - including some plots of the target/decoy curves (e.g. such as those in Rosenberger et al 2017 Nat Methods) would also strengthen this argument that DIA-NN is properly controlling FDR.

- Fig 4D profiles of log2 fold changes would benefit from error bars.

Reviewer #2 (Remarks to the Author):

The manuscript from Steger et al. describes improvements to the methods used to characterize site specific protein ubiquitination. Key advances in the paper include an optimized lysis protocol, a data independent analysis analytical method and a data processing approach that takes advantage of neural networks. The authors employed their optimized DIA—MS method to characterize the ubiquitination events modulated by inhibitors of the deubiquitinase USP7. There, the authors compared DIA acquired DiGly data across four distinct small molecule inhibitors of USP7. Overall, many elements of this manuscript are convincing enough to say that the method works, and may even work very well. That said, there is one aspect of this protocol where there is the potential for a dramatic artifactual result, that if true, would profoundly alter conclusions about the comparative effectiveness of this method versus others. This is detailed below and will need to be addressed rigorously. On the USP7 substrate identification side, the results are overall interesting and impressive. The authors are commended for recognizing the valuable insights that come from being able to “accurately time ubiquitination and consequent changes in [protein] abundance. Acute treatments are key, and leveraging them herein brings rigor to the biology side of this story There also, an experiment that could be done which would markedly enhance the impact of the paper and that the authors are greatly encouraged to execute during revision.

As noted, the biological experiments performed using proteasome and DUB inhibitors make it clear that the method described works to identify DiGly peptides. That said, a major premise of the paper is that the improved SDC based lysis protocol makes this method far superior to existing methods. That need not be the case for this paper to be acceptable for publication, but as written, it must be validated. In terms of validation, the authors have provided the perfunctory comparisons of numbers of identified peptides, %CV, etc. That said, all of these numbers could be dramatically affected by the possibility that the method is creating large numbers of artifactual peptides with isobaric 114.0429 Da mass additions that are nearly indistinguishable from DiGly modifications. Mann and colleagues previously reported this concern with regards to iodoacetamine (Nielsen et al.

Nat Meth 2008), and in that paper proposed the use of chloroacetamide as was used here. In their paper, Mann and colleagues did note the potential for CAA (as used here) to generate the same artifactual DiGly-mimicking modifications if incubated at high temperature. That is precisely what was done here in the novel SDC lysis protocol, with 40 mM CAA incubation performed at 95C for 10 min. To evaluate this, the authors should provide a systematic assessment of their SDC lysis protocol (+/- CAA treatment at 95C and lower temps), looking not in the DiGly enriched samples...but rather in the digested whole cell lysate. It remains possible that the profound increase in DiGly peptide sensitivity comes from the generation of a huge pool of artifactual features and the authors must show that this is not the case.

As an aside, one way one can diagnose this will be to search unenriched digested lysates. If the DIA-MS method is identifying any more than a handful of modified peptides (i.e. K48-Ub, K119-H2A), the artifact is a major issue. A confirmatory way to evaluate this is to look for peak doublets for features such as the K48 and K63 modified peptides from Ub in DiGly enriched samples. The bona fide and artifactual 114.0429 Da modified peptides nicely resolve on most every LC-MS setup.

What happens to DiGly peptide number is CAA treatment is performed at lower temperatures?

Authors should show some form of raw abundance data for the DiGly peptides of Ub and their abundances under MG132 and USP7-inhibitor treated conditions as a demonstration to the reader. A disconcerting thing about DIA data is the amount of processing that must be done blindly without the user having an opportunity to look in at the data. These sorts of demonstrations will help to provide confidence in the analytical assertions being made.

In Suppl. Note 1, the authors describe an E.Coli mixing experiment. This would be an ideal case to test the CAA artifactual modification question since these bacterial do not express Ub. If DiGly peptides are identified in these samples prepared using the SDC lysis protocol, that would be clear evidence of the effect.

Rather than showing peptide counts, can the authors provide data in the early figures to show the extent of increased abundances for individual DiGly peptides upon proteasome inhibition. Options would be histograms, bees warm dot plots, etc. A key question will be how many do not change, or go down...some certainly do as a result of Ub depletion and it would build confidence to see that those go down (e.g. K119-H2A).

The other major point the authors should address relates to the specific versus off target effects of USP7 small molecule inhibitors. In the event that USP7 inhibitors are working through targets

besides the DUB of interest, those would be effects that would be visible in USP7 KO or knockdown cells. In other words, if the target is absent but the drug still has an acute effect, it makes clear that the compound is working through another enzyme. Moreover, the DUB KO or KD should also independently confirm many of the targets identified here, acknowledging that compensation will also complicate results of this experiment. Addition of a USP7 KO experiment would be a capstone piece of data for his paper.

The term 'ubiquitinomics' feels awkward...is there any other term that can be used to describe this area of work.

On p9, the text states that "due to cytotoxicity. Proteasome inhibitors cannot be utilized for studying protein ubiquitination in vivo". This is untrue. Molecules such as bortezomib and carfilzomid are clinically approved molecules in spite of their toxicities, and indeed hit their targets in vivo. This statement should be revised to ensure that it remains factually accurate, even while conveying the intended message.

The very next sentence begins "We therefore tested...". This is a logical disconnect and does not make sense relative to the text that comes before it.

Overall, I did not find the figures overly informative due to the fact that the majority show high level overviews of the data, In addition to these, or in place of certain elements, it would be comforting to see more 'raw' data for key positive control features such as known USP7 substrates. Chromatographic traces convey how clean or noisy data are in a way that is often obscured when the data is aggregated as they are here and in other such papers. The authors have a chance to make a major impact, and this is a key piece that would help do so. It would be great to do so for some of the features so elegantly shown in Fig 4C

Lastly, the one thing that seems missing from this paper is any text describing how the Neural Networks analysis method operates or how/why it is better. This was a major tenet of impact in the early text, but was not spoken about in any detail, nor were any demonstrative figures shown.

Figure S7 has the y-axis value truncated in a way that obscures the true result.

On p29, the methods describe two different STAGE tip protocols. When were each used? Are there any key performance differences between them that might affect data interpretation?

Reviewer #3 (Remarks to the Author):

The authors describe here a novel workflow for ubiquitomics, the analysis of ubiquitylated proteins in a biological sample. This workflow is based on a sample preparation protocol using an SDC (sodium deoxycholate)-based cell lysis, prior to the enrichment of the modification, followed by data-independent acquisition (DIA) coupled to a neural network-based data processing.

This methodology increases significantly the sensitivity and the throughput of the current GlyGly peptidome analysis workflow. Finally, the authors applied this workflow to characterise four different USP7 inhibitors in a colon cancer cell line (HCT116).

General comments:

- The manuscript is well written and clearly presented. However, the use of the English language could be improved.
- The experiments are well designed and the quality of the figures is good.
- This study is of high interest to other researchers in the field. However, a very similar approach, SDC-based cell lysis followed by DIA analysis) has been published recently in this journal: Hansen, et al., 2020, <https://doi.org/10.1038/s41467-020-20509-1>. This takes in part the novelty of this study off.

Major points:

- Lack of orthogonal validation of the described workflow using more basic biochemistry techniques to perform ubiquitylation assays. For example, doing GST-UBA pulldowns as in Hansen, et al., 2020 (Fig S3) or performing Ubiquitin immunoprecipitations as in Akimov et al., 2018 (<https://doi.org/10.1038/s41594-018-0084-y>; Figure 5). This would be particularly interesting for validating some of the USP7-dependent identified GG peptides.
- The acquisition methods (LC-MS/MS measurements) are not comparable for DDA (more conservative) and DIA (more relaxed) and this could lead to different sensitivities. For instance, I consider the average of 4 data points low (8-10 data points would be my recommendation). Table S8 describing the optimisation of the number of DIA segments is missing.

- ~10% of the GG peptides identified by DDA do not overlap with the GG peptides identified by DIA. How would you explain this if the DIA method is more sensitive?

Minor points:

- The authors use Usp7 to refer to the human protein. The right nomenclature is USP7 and this should be amended.

- The authors mention that simultaneously recording the dynamics of both proteome and ubiquitome is a strategy for capturing USP7 targets or substrates (i.e. in the abstract or on page 11). This is not correct since these techniques do not provide evidence for direct interaction and we cannot rule out that the observed effects might be a consequence of secondary ubiquitylation events. The should be more carefully reformulated across the manuscript.

- deep neural network-based data processing is mentioned several times across the text. I brief description of such an approach would be useful for gaining access to a broader audience.

Point-by-point response

Summary

We thank all Reviewers for their effort in reading our manuscript and for their thorough assessment of our work, as well as for their insightful suggestions on how to improve it. We have now significantly reworked the manuscript and have addressed all the concerns raised. The main changes of the paper are as follows:

- We have included an experiment, showing that CAA treatment at high temperatures does not cause di-carbamidomethylation of lysine residues (Supplementary Figure 1). We provide two additional proofs in the underlying response, demonstrating that CAA does not induce di-carbamidomethylation of lysines.
- We further optimised DIA-NN, substantially increasing the numbers of ubiquitinated peptides identified and quantified. We reprocessed all data with this new software version and with modified search parameters (maximum of 2 variable modifications/peptide instead of just one, as used in the original manuscript). We chose these parameters to more closely reflect the standard settings of MaxQuant and Spectronaut. To make our data better comparable to other studies (such as the study by Hansen et al, see below), we also adjusted the counting of K-GG peptides. In the original manuscript, we counted modified sequences (e.g. $XXM(O_x)XK(GG)X$ and $XXMXK(GG)X$ would be counted twice, where X is any amino acid), whereas in the revised manuscript, we count K-GG stripped sequences (the abovementioned peptide would be counted just once).
- We also included several benchmark experiments (Figures 2, 3 and Supplementary Figure 8) to demonstrate that DIA-NN significantly outperforms the alternative state-of-the art DIA data processing software (Spectronaut) for analyzing ubiquitinomics raw data, at a comparably low false discovery rate.
- We reworked the figures to make them more informative and revised the wording of the text. We also performed several biological validation experiments. First, by reducing expression of USP7 using siRNAs, we demonstrate that the observed USP7 inhibitor effects are on-target (Figure 4 and Supplementary Figure 5). Second, we validate USP7 auto-regulation biochemically using ubiquitin-binding domain pulldown and Western blotting as well as MS. Finally, we likewise use ubiquitin-binding domain pulldown coupled to MS to validate the ubiquitination (upon USP7 inhibition) of the identified USP7 targets (Supplementary Figure 6).
- To compare the different USP7 inhibitors in a more unbiased manner, we did not include any fold-change cut-off in the revised manuscript (originally >2-fold) (see Figure 5 in both the

original and the revised manuscripts). As a result, the numbers of the correlation plot and the Venn diagram changed slightly. However, the main conclusions are not affected in any way.

Reviewer #1 (Remarks to the Author):

The authors of this manuscript, “Time-resolved in vivo ubiquitinome profiling by DIA-MS reveals USP7 targets on a proteome-wide scale”, present a novel mass spectrometry-based approach for studying ubiquitinated proteins, and apply their approach to study the effects of several inhibitors of the deubiquitinase USP7 on protein degradation. The authors sufficiently demonstrate the improvements made by their proposed ubiquitinome protocol,

Thank you for your positive feedback!

although unfortunately much of the novelty of using SDC and DIA has been recently “scooped” by the Hansen et al 2021 Nat Comms paper published in this journal.

We note that we posted a preprint of our work back-to-back with the study of Matthias Mann’s laboratory on bioRxiv back in July 2020

(<https://www.biorxiv.org/content/10.1101/2020.07.24.219055v1>,

<https://www.biorxiv.org/content/10.1101/2020.07.23.218651v1>).

That said, there is quite a bit of informative, but downplayed, data on the implications of using DIA versus DDA that might be expanded to regain some novelty in that regard.

We reworded the text to make it clearer that the potential of DIA over DDA is huge for ubiquitinomics applications. In general, we significantly improved the manuscript to clearly show that our workflow outperforms other already peer-reviewed methods.

Overall, the authors seem quite knowledgeable of the mass spectrometry-based ubiquitination field and have presented interesting datasets. I imagine the manuscript’s conclusions will be of interest to mass spectrometrists performing ubiquitin PTM work, and perhaps also assist drug discovery efforts geared towards degraders such as the ubiquitin-proteasome pathway.

I recommend publication of the manuscript with the following major and minor revisions to support the interpretation and the presentation of the manuscript’s main points.

Thank you!

Major issues:

(1) Although the authors do mention the Hansen et al preprint in the Discussion, that work has now been published in this journal (Hansen et al 2021, Nat Comms). As the Hansen paper also used SDC and DIA for ubiquitination, the authors should adjust their language appropriately (e.g. use of “novel” is already a bit dated, as Hansen et al did it “first”).

Response 1: We acknowledge that the manuscript from Hansen et al got peer reviewed and published much faster than ours and we now included the Nat Comms citation also in the introduction (present in the discussion in the original version). Although we posted our work back-to-back on bioRxiv, as stated above, we adjusted the language slightly (e.g. we omit statements like novel protocol). For example, see abstract (Page 2, lines 4 and 5).

However, there is much room to compare/contrast the salient points of this work versus Hansen et al. As one example, Hansen et al relies on Spectronaut for DIA analysis, which is a closed-source and proprietary software, whereas the authors here are using a free and open-source software, DIA-NN. A few more detailed points below:

(1a) Although the authors have done well to benchmark DIA-NN for ubiquitin modifications, there is no comparison of DIA-NN versus other software. I would like to see a comparison of DIA-NN versus Spectronaut (for the sake of comparing to Hansen et al) or, if a Spectronaut license is not possible, some other DIA software like IPF (Rosenberger et al 2017, Nat Biotech) or Thesaurus (Searle et al 2019, Nat Methods).

Response 2: We agree that the quality of our paper will benefit from a side-by-side comparison of DIA-NN with another frequently used software for DIA data analysis. To do so, we first processed 6 raw files from Hansen et al with the library-free setup in DIA-NN. Specifically, the samples were derived from HEK293 cells treated with 10 μ M MG-132 for 4 h and comprised three workflow replicates (2 injection replicates for each of those, see Figure 1d, Hansen et al, <https://doi.org/10.1038/s41467-020-20509-1>). This yielded an increase in K-GG remnant peptides of more than 40% (Figure 2g of the revised manuscript).

We also used another set of raw files from Hansen et al (U2OS cells, DMSO-treated, Figure 3 of Hansen et al) and processed them with the library-free search option in DIA-NN. Strikingly, in this direct comparison, DIA-NN again quantified about 75% more K-GG peptides (Figure 3e of the revised manuscript).

Finally, we processed our E.coli/human K-GG data set with Spectronaut, showing that DIA-NN keeps the false discovery rate low (comparable or better than MaxQuant and Spectronaut), while identifying 75% more human K-GG peptides than Spectronaut and about four times more than MaxQuant (Supplementary Figure 8 of the revised manuscript).

(1b) The authors claim novelty with a library-free DIA performing comparably to DIA analysis based on experimentally-generated spectral libraries, e.g. in the Methods, “Briefly, DIA-NN first generates a spectral library from DIA data using all identifications in specified raw files.” I would like a bit more elaboration on this point, as to my knowledge none of the other predicted spectral library algorithms (MS2PIP, Prosit) can generate modified peptide spectra. Does DIA-NN work for other modifications as well? Does it predict retention times, like other predicted spectral library tools? What about multiply-modified peptides?

Response 3: Indeed, prediction of modified peptide spectra has been lagging behind. Only recently it was successfully applied using a number of different algorithms. The most general way to implement this is to consider modified amino acids (e.g. K-GG or phospho-Ser) as 21st, 22nd, ... types of amino acids. K-GG would therefore be simply treated as unmodified amino acids, and the deep learning algorithm would be oblivious to its relation to a lysine. This idea is discussed, for example, in a recently published paper by the Cox group (DeepMass:Prism, <https://www.nature.com/articles/s41592-019-0427-6>). Another way is to encode each residue in the peptide as a pair of numbers - one for the amino acid, and another one for the modification (zero if no modification is present). This approach is used in pDeep2 and pDeep3, which are publicly available tools for modified peptide spectra prediction (<https://pubs.acs.org/doi/abs/10.1021/acs.analchem.9b01262>, <https://www.biorxiv.org/content/10.1101/2020.09.13.295105v1>). In both cases, an arbitrary number of modified residues is theoretically supported, although the prediction performance might vary. It appears that Spectronaut (starting from version 14) likewise uses deep learning for spectra prediction, although its method is undisclosed and unpublished.

The module that is currently used for modified peptide prediction (spectra, retention times and ion mobilities) in DIA-NN (since Jan 2020) is not novel idea-wise, therefore we haven't published it. However, there are a number of technical optimisations in it, which we intend to publish in the future as part of a larger project. It aims at enabling high quality prediction of spectra, RTs and ion mobilities for peptides with arbitrary modifications.

That said, we plan the next major release of DIA-NN for July (that is, before the present paper will be published), and it will come with extensive documentation describing all its features, including prediction of modified peptide spectra by deep learning. Apart from K-GG, variable modifications supported by DIA-NN's deep learning predictor are M(ox), N-term acetylation, phospho (STY) and deamidation (NQ), and DIA-NN supports any number of modified residues per peptide. We consider these modifications beyond the scope of this work, but do plan to benchmark them in future works.

Of note, DIA-NN also features a more simplistic predictor (introduced previously <https://www.nature.com/articles/s41592-019-0638-x>) based on MS Simulator, which can be trained by

the user to predict y-series fragmentation and retention times for peptides bearing arbitrary modifications. This functionality is not used in the present work but all relevant documentation can be found in the manual of DIA-NN.

In summary, we believe that with respect to data processing, the novelty and the strength presented in here, are the combination of previously described features of DIA-NN (primarily the excellent performance of the neural network classifier, especially in library-free mode <https://www.nature.com/articles/s41592-019-0638-x>, Supp. Fig. SN8.2) and a new PTM-scoring module (described in Methods of the current work). We demonstrate that this combination clearly outperforms the existing state-of-the-art software (Spectronaut) for ubiquitinome analyses. Once online, we will suggest the present paper as reference for DIA-NN users who are interested in PTM data processing.

(2) The authors use an unusual type of plot in Figures 4B and S3. These types of figures are usually shown as volcano plots, but here the author is using “standard error” instead of “p-value” on the vertical axis. I request the authors provide volcano plots for these figures, at least as supplemental figures, because these plots are so unusual for proteomics data.

Response 4: We agree that the scientific community is used to seeing volcano plots with p-values instead of standard errors on the y-axis. We changed the plots in Figure 4b and Supplementary Figures 3 and 7 accordingly. Thank you for this suggestion!

Minor points:

- Pg 3 paragraph 2, “For example, while K11- and K48-linked chains can tag proteins for proteasomal degradation, ubiquitin conjugates on K63 tend to modulate protein-protein interactions”. It’s unclear if this refers to some specific protein with ubiquitinated K11, K48, and K63 residues, or if this means that the lysine can have 11-, 48-, or 63-ubiquitin moiety-long chains on a lysine.

Response 5: We changed this statement to make it clear that proteins modified with K63-linked ubiquitin chains modulate protein-protein interactions (Page 3, line numbers 18-19). (*...For example, while K11- and K48-linked chains can tag proteins for proteasomal degradation, attachment of K63-linked ubiquitin conjugates to target proteins tends to modulate protein-protein interactions.*)

- Pg 3, paragraph 4, “By introducing an SDC-based lysis protocol [...]” the abbreviation “SDC” should be defined first before using the abbreviation.

Response 6: Thanks for spotting this inconsistency! We have changed it to: ...a sodium deoxycholate (SDC)-based...(Page 3, line number 26).

- Pg 3, paragraph 4, “[...] as well as significantly improve quantitative accuracy.” The authors show quantitative precision/reproducibility, but only show better accuracy in Supplementary Figure 7 which is actually a very interesting figure that isn’t mentioned at all in the text! Authors should use some text to describe that DIA improves quantitative accuracy per Fig S7.

Response 7: We fully agree that this is a very important finding and this should be better highlighted. On page 7, line numbers 11-12, we mention that DIA-NN provides excellent quantitative accuracy and we refer to supplementary note 2. We now additionally refer to Supplementary Figure 9 (which was Figure S7 in the originally submitted manuscript) to make it easier for the reader to locate the data.

- Fig 1B - on average, how many lysine residues are ubiquitin-modified per protein? One ubiquitinated lysine per protein? Two? Six?

Response 8: There are 6 modified lysine residues per protein on average and we have included this in the main text of the revised manuscript (Page 4, line numbers 22-23), *In total, we identified 41,094 K-GG remnant peptides and on average found 6 –GG modified lysine residues per protein.*

- Pg 7, paragraph 2, “Although DIA-NN uses a rigorous approach to determine the false discovery rate (FDR) [...]” - including some plots of the target/decoy curves (e.g. such as those in Rosenberger et al 2017 Nat Methods) would also strengthen this argument that DIA-NN is properly controlling FDR.

Response 9: We routinely include FDR-validating plots in our works, and DIA-NN’s performance with regards to identification confidence has been validated already multiple times and on different types of data, clearly demonstrating its reported FDR estimates are actually very conservative (<https://www.nature.com/articles/s41592-019-0638-x>, <https://www.nature.com/articles/s41587-021-00860-4>, <https://www.biorxiv.org/content/10.1101/2021.03.08.434385v1>). To our knowledge, no other DIA software has been validated that strictly. For example, Spectronaut has not been validated for false discovery rate of modified peptide detection. Rather, the work by Bekker-Jensen et al. has established that the false localisation rate of phosphosites is controlled at about 3%, on a selected set of spike-in peptides (<https://doi.org/10.1038/s41467-020-14609-1>). Similarly, Thesaurus (part of EncyclopeDIA) has also been shown to control false localisation rate at 7% (<https://doi.org/10.1038/s41592-019-0498-4>).

In the present work, however, we refrain from claiming that the FDR estimates for modified peptides are exact, but rather demonstrate that the FDR is low and well-controlled (at about ~1%, when filtered at 1% q-value).

The primary reason for this is that in a ‘real’ experiment featuring a K-GG-enriched human sample, the effective FDR is likely to be significantly lower than what we have estimated for our K-GG-enriched human + non-enriched *E.coli* benchmark experiment (see Supplementary Figure 8). That is, the human-*E.coli* benchmark is conservative and provides an upper FDR estimate.

Indeed, if we exaggerate and assume that every single lysine in the sample bears a -GG adduct, then there would not be any need for an extra PTM-specific scoring module. The problem arises when unmodified peptides are present in the sample, which is the case for ubiquitinomics. These can be mistaken as modified peptides and need to be spotted by the PTM-scoring module. In DIA-NN, this is implemented in such a way as to provide conservative estimates regardless of whether unmodified peptides are present, by relying, to a significant extent, on direct MS2-level evidence for fragment ions containing the modified residue(s). That’s why it performs very well even in the very strict human-*E.coli* benchmark experiment. We suspect that the implementation in Spectronaut is similar (as it also controls the FDR well in our *E.coli*/human benchmark experiment).

- Fig 4D profiles of log2 fold changes would benefit from error bars.

Response 10: Makes total sense, we included those in the revised manuscript (Figure 4d). Thank you for the suggestion!

Reviewer #2 (Remarks to the Author):

The manuscript from Steger et al. describes improvements to the methods used to characterize site specific protein ubiquitination. Key advances in the paper include an optimized lysis protocol, a data independent analysis analytical method and a data processing approach. that takes advantage of neural networks. The authors employed their optimized DIA—MS method to characterize the ubiquitination events modulated by inhibitors of the deubiquitinase USP7. There, the authors compared DIA acquired DiGly data across four distinct small molecule inhibitors of USP7. Overall, many elements of this manuscript are convincing enough to say that the method works, and may even work very well.

Thank you very much for your positive feedback!

That said, there is one aspect of this protocol where there is the potential for a dramatic artifactual result, that if true, would profoundly alter conclusions about the comparative effectiveness of this method versus others. This is detailed below and will need to be addressed rigorously.

This is a very valid concern and we addressed this with three experiments. We suggest including only one of those experiments in the manuscript (see details below).

On the USP7 substrate identification side, the results are overall interesting and impressive.

Thank you!

The authors are commended for recognizing the valuable insights that come from being able to “accurately time ubiquitination and consequent changes in [protein] abundance. Acute treatments are key, and leveraging them herein brings rigor to the biology side of this story. There also, an experiment that could be done which would markedly enhance the impact of the paper and that the authors are greatly encouraged to execute during revision.

We provide one validation experiment using siRNA-mediated knock-down of USP7 (see details below).

As noted, the biological experiments performed using proteasome and DUB inhibitors make it clear that the method described works to identify DiGly peptides. That said, a major premise of the paper is that the improved SDC based lysis protocol makes this method far superior to existing methods. That need not be the case for this paper to be acceptable for publication, but as written, it must be validated.

We now provide the requested validation experiments, please see below.

In terms of validation, the authors have provided the perfunctory comparisons of numbers of identified peptides, %CV, etc. That said, all of these numbers could be dramatically affected by the possibility that the method is creating large numbers of artifactual peptides with isobaric 114.0429 Da mass additions that are nearly indistinguishable from DiGly modifications. Mann and colleagues previously reported this concern with regards to iodoacetamine (Nielsen et al. Nat Meth 2008), and in that paper proposed the use of chloroacetamide as was used here. In their paper, Mann and colleagues did note the potential for CAA (as used here) to generate the same artifactual DiGly-mimicking modifications if incubated at high temperature. That is precisely what was done here in the novel SDC lysis protocol, with 40 mM CAA incubation performed at 95C for 10 min.

Response 11: Matthias Mann and colleagues indeed reported that IAA can induce di-carbamidomethylation of lysine residues, resulting in the addition of a 114.0429 Da mass tag that is indistinguishable from a K-GG adduct in terms of added mass to the peptide. Mann and colleagues however failed to detect di-carbamidomethylation of lysines when treating peptides with CAA at room temperature. Nevertheless, one can speculate that CAA treatment, along with incubation at high temperatures, could induce di-carbamidomethylation of lysines. However, the chemical structures of di-glycine lysine and di-carbamidomethylated lysine largely differ from each other (Figure 1 of this

response). Thus, it is very unlikely that the monoclonal K-GG antibodies used for enrichment of ubiquitin remnant peptides would recognize a di-carbamidomethylated lysine residue. Nonetheless, it is possible that a small fraction of peptides with di-carbamidomethylated lysines would be non-specifically bound to the sepharose matrix used for immunoprecipitation of K-GG peptides. We therefore performed three experiments to exclude that CAA treatment at high temperatures causes an increase of MS-detectable peptides containing K residues with a mass shift of 114.0249 Da (see below).

Response Figure 1: Chemical structures of a (a) di-carbamidomethylated and (b) -GG-modified lysine residue

To evaluate this, the authors should provide a systematic assessment of their SDC lysis protocol (+/- CAA treatment at 95C and lower temps), looking not in the DiGly enriched samples...but rather in the digested whole cell lysate. It remains possible that the profound increase in DiGly peptide sensitivity comes from the generation of a huge pool of artifactual features and the authors must show that this is not the case.

Response 12: Thank you for actively proposing an experiment for addressing this potential issue. As suggested, to verify this, we lysed HCT116 cells in SDC buffer without CAA, followed by heating to 95°C. After cooling to room temperature, we added 40 mM CAA to all samples and then incubated them at 35/50/65/80/95°C for 10 min. We then processed the samples for whole proteome MS analysis and acquired the data in DDA, followed by searching the resulting raw files with MaxQuant (standard settings, inclusion of K-GG as variable modification, match-between-runs disabled). This revealed that only a handful of K-GG peptides were identified in all conditions (well within the expected false discovery rate of the processing software) and that there was no temperature effect.

We present this data in Supplementary Figure 1A of the revised manuscript. We revised the corresponding section in the main text accordingly (Page 4, starting from line number 13).

As an aside, one way one can diagnose this will be to search unenriched digested lysates. If the DIA-MS method is identifying any more than a handful of modified peptides (I.e. K48-Ub,

K119-H2A), the artifact is a major issue. A confirmatory way to evaluate this is to look for peak doublets for features such as the K48 and K63 modified peptides from Ub in DiGly enriched samples. The bona fide and artifactual 114.0429 Da modified peptides nicely resolve on most every LC-MS setup.

Response 13: Thank you for proposing another experiment. We searched a non-enriched proteome raw file (proteome samples shown in Figure 4, i.e. USP7i time course) with K-GG as variable modification (library-free mode in DIA-NN). These samples were lysed with SDC buffer (including 40 mM CAA, heated to 95°C for 10 min) as described in the methods section. As rightly pointed out by the referee, there should be little, if any at all, K-GG peptides detectable in such an analysis; hence, some of the identified K-GG peptide most likely represent either a false positive identification or an artificially introduced di-carbamidomethylation (114.0249 Da) on a lysine residue. This revealed only 275 K-GG peptide precursors, in contrast to the 144,200 precursors quantified in total (Figure 2 of this response). Besides validating that the false discovery rate for K-GG peptides is well below 1% (since $275 / 144,200 = 0.002$), we conclude that despite the high concentration of CAA in the lysis buffer and the heating step at 95°C, there is no significant artificial di-carbamidomethylation (114.0249 Da) of lysine residues detectable.

Response Figure 2: K-GG peptide precursors identified in a whole proteome tryptic digest from HCT116 cells lysed with SDC buffer

HCT116 cells were lysed with SDC buffer (1% SDC, 10 mM TCEP, 40 mM CAA, 75 mM Tris-HCl) and heated at 95°C for 10 min. The proteins were digested using trypsin and the resulting peptides desalted. 800 ng of the digest were injected into the mass spectrometer and the data acquired in DIA mode using a 125 min LC gradient. The data was processed with DIA-NN, employing a library-free search strategy with deep learning for prediction of retention times and peptide spectra. K-GG was

included as variable modification. This revealed 275 peptide precursors harboring a K-GG modification, in contrast to the 143,925 unmodified peptides detected (that is only 0.2% of all peptides reported were K-GG).

What happens to DiGly peptide number if CAA treatment is performed at lower temperatures?

Response 14: Thank you for the suggestion. However, we think that the results obtained with such a setup would be inconclusive. For cell lysis in SDC buffer, cells must be heated to high temperatures and if we did not add CAA to the buffer, DUBs would be inhibited less efficiently. Consequently, SDC buffer containing CAA would possibly identify more K-GG peptides, but this would not necessarily be an artifact but rather the inhibitory effect of CAA towards DUBs. We therefore did not perform this experiment, but hope that the two experiments described earlier in this response, as well as an additional experiment described below (Response Figure 3), provide convincing evidence that CAA treatment at high temperatures does not introduce artificial di-carbamidomethyl-lysine modifications.

Authors should show some form of raw abundance data for the DiGly peptides of Ub and their abundances under MG132 and USP7-inhibitor treated conditions as a demonstration to the reader. A disconcerting thing about DIA data is the amount of processing that must be done blindly without the user having an opportunity to look in at the data. These sorts of demonstrations will help to provide confidence in the analytical assertions being made.

Response 15: K-GG peptides derived from Ubiquitin are highly abundant and are among the most strongly enriched peptides by the K-GG-specific antibodies. Upon MG-132 treatment, their signal is saturated in the MS measurements. To quantify K-GG peptides derived from Ubiquitin, one would need to perform either a SILAC experiment (similar to Akimov et al, doi: 10.1038/s41594-018-0084-y) or use isotopically labeled standards (see Xu et al, doi: 10.1016/j.cell.2009.01.041). In both of these publications, it was demonstrated that MG-132 treatment leads to an increase of all Ubiquitin chain types. We therefore consider this experiment outside the scope of this study.

However, we included raw data for several experiments in the revised manuscript. For example, the bee swarm dot plot in Figure 3B shows Log₂ fold changes of MG-132 vs DMSO treated cells. It demonstrates that most detected K-GG peptides are upregulated upon MG-132. However, some peptide quantities are downregulated upon MG-132 treatment, including H2A-K119. This site along with other control sites are highlighted in the plot.

Additionally, we show protein abundances for targets identified in the ubiquitin domain pulldown experiment, which is presented in Supplementary Figure 6 of the revised manuscript. We included a set of supplementary tables (see Supplementary Data 1-11 for the different experiment), in which we comprehensively included most raw data acquired for this study.

In Suppl. Note 1, the authors describe an E.Coli mixing experiment. This would be an ideal case to test the CAA artifactual modification question since these bacterial do not express Ub. If DiGly peptides are identified in these samples prepared using the SDC lysis protocol, that would be clear evidence of the effect.

Response 15: We performed the following experiment: using commercially available tryptic digest from E.coli, we either resuspended the peptides in 0.1% formic acid or in SDC buffer. We then heated the SDC-solubilized peptides to 95°C for 10 minutes. Following desalting, resuspension in 0.1% formic acid and concentration estimation, we injected 800 ng of SDC- and formic acid-solubilized peptides into the mass spectrometer (4 injection replicates each), acquiring the data in DDA mode. MaxQuant quantified about 9,000 E.coli peptides in each sample and found 11 and 10 K-GG peptides on average, for SDC-lysed and formic acid-solubilized peptides, respectively. Stringent filtering of the data (posterior error probability (PEP)<0.001) drastically reduced the K-GG IDs (Figure 3 of this response). Also in this case, we conclude that CAA treatment at 95°C does not introduce artificial K-GG mimicking peptides.

Response Figure 3: K-GG peptides identified in a tryptic digest of E. coli

10 µg of a tryptic digest from E. coli (Waters (186003196)) were resuspended in either 0.1% formic acid or SDC buffer (including 40 mM CAA). The SDC lysate was further processed by heating it to 95°C for 10 minutes. Subsequently, peptides in the SDC buffer were desalted. Both peptide pools were then injected into the MS and the data acquired in DDA mode. The database search was done using MaxQuant (standard settings, no match between runs, K-GG as variable modification). K-GG and total peptides were counted using the 'modification specific peptide' output from MaxQuant. The left bar plot shows all quantified E.coli peptides, whereas the plot on the right shows K-GG peptides with posterior error probability (PEP)< 0.001. SDC= sodium deoxycholate, FA= formic acid.

Rather than showing peptide counts, can the authors provide data in the early figures to show the extent of increased abundances for individual DiGly peptides upon proteasome inhibition. Options would be histograms, bees warm dot plots, etc. A key question will be how many do not change, or go down...some certainly do as a result of Ub depletion and it would build confidence to see that those go down (e.g. K119-H2A).

Response 16: Thank you for the suggestion. We show a bee swarm dot plot of individual log2 fold changes (MG132 vs DMSO) of K-GG peptides. This revealed that the majority of peptide intensities increase upon MG-132 treatment. However, there are peptides that decrease, and those include K119-H2A. We present this data in Figure 3B of the revised manuscript and modified the main text accordingly (see Page 9, starting from line number 34).

The other major point the authors should address relates to the specific versus off target effects of USP7 small molecule inhibitors. In the event that USP7 inhibitors are working through targets besides the DUB of interest, those would be effects that would be visible in USP7 KO or knockdown cells. IN other words, if the target is absent but the drug still has an acute effect, it makes clear that the compound is working through another enzyme. Moreover, the DUB KO or KD should also independently confirm many of the targets identified here, acknowledging that compensation will also complicate results of this experiment. Addition of a USP7 KO experiment would be a capstone piece of tdata for his paper.

Response 17: Although we have shown that four structurally distinct inhibitors targeting USP7 have similar effects (Figure 5 of the submitted manuscript), we now included an additional validation experiment showing that FT671 is a selective USP7 inhibitor. We down-regulated USP7 with 2 different pools of siRNAs. After 48 h of knock-down, we achieved an approximately 4-fold downregulation of the protein with both siRNAs (Supplementary Figure 5 of the revised manuscript). We quantified both the proteome and the ubiquitinome after DMSO or FT671 treatment (5 minutes), for siCTRL and siUSP7 samples, to a depth of 10,000 proteins and 40,000 K-GG peptides in each measurement. Analysis of the proteome revealed drastic changes in global protein expression upon USP7 knockdown, including a significant stabilization of p53 with both siRNAs, as expected (see Supplementary Figure 5 and Supplementary Data 9 of the manuscript).

In the ubiquitinome, we found an overall strongly decreased degree of regulation after compound treatment when USP7 expression was reduced using siRNAs. In fact, compared to cells transfected with a control siRNA, in siUSP7-treated cells, ubiquitination sites of virtually all putative USP7 substrates pinpointed in this study were induced less strongly upon compound treatment. We included this data in Figure 4 of the revised manuscript and modified the text (Page 13, starting from line 25).

The term ‘ubiquitinomics’ feels awkward...is there any other term that can be used to describe this area of work.

Response 18: We are not aware of any alternative term for ubiquitinomics. We tried to change the language slightly and to replace ubiquitinomics with MS-based ubiquitin profiling, whenever appropriate.

On p9, the text states that “due to cytotoxicity. Proteasome inhibitors cannot be utilized for studying protein ubiquitination in vivo”. This is untrue. Molecules such as bortezomib and carfilzomid are clinically approved molecules in spite of their toxicities, and indeed hit their targets in vivo. This statement should be revised to ensure that it remains factually accurate, even while conveying the intended message.

Response 19: It is true that proteasome inhibitors are used for cancer therapy and to avoid confusion, we removed the sentence from the text (Page 9, line number 25).

The very next sentence begins “We therefore tested...”. This is a logical disconnect and does not make sense relative to the text that comes before it.

Response 20: Thank you for the careful assessment. We have changed the sentence to as follows (Page 9, line 25):

To determine whether our DIA-MS method was suited for high coverage ubiquitinomics in the physiological context and in case of low protein input, we quantified K-GG peptides derived from different protein input amounts of either MG-132- or DMSO-treated cells.

Overall, I did not find the figures overly informative due to the fact that the majority show high level overviews of the data, In addition to these, or in place of certain elements, it would be comforting to see more ‘raw’ data for key positive control features such as known USP7 substrates.

Chromatographic traces convey how clean or noisy data are in a way that is often obscured when the data is aggregated as they are here and in other such papers. The authors have a chance to make a major impact, and this is a key piece that would help do so. It would be great to do so for some of the features so elegantly shown in Fig 4C

Response 21: In this study, we have generated an overwhelming amount of data and have tried to give the best overview on the data in the main figures. For transparency, we have uploaded all raw data and processings (MaxQuant and DIA-NN) to ProteomeXchange and everyone interested in using or reproducing our data is welcome to do so. Accession details can be found in the methods section. We have also included more supplementary tables in the revised manuscript, reporting raw intensities, and

p-values and fold changes, when appropriate (see Supplementary Data 1-11). That said, we have reworked the figures to make them visually more appealing and included more positive controls, such as the K-GG sites of H2A in Figure 3b or the quantifications of different proteins upon ubiquitin domain pulldown experiments (Supplementary Figure 6).

To give the best overview of what we think are bona-fide USP7 substrates targeted for proteasomal degradation, we presented an aggregated heat map with mean Log2 fold changes of downregulated proteins and their corresponding ubi profiles (averaged signal of significantly-upregulated features. Figure 4c). The presented comparison of both the proteome and the ubiquitinome are statistically significant observations. To make the data better accessible to the reader, we included an additional table (Supplementary Data 8), consisting of a list of proteins and their corresponding ubi sites shown in the heatmap. In the table, we included raw intensities, fold-changes and p-values of all features. We hope that this addresses their concern.

Lastly, the one thing that seems missing from this paper is any text describing how the Neural Networks analysis method operates or how/why it is better. This was a major tenet of impact in the early text, but was not spoken about in any detail, nor were any demonstrative figures shown.

Response 22: We have thoroughly described and benchmarked the neural network classifier in the DIA-NN paper <https://www.nature.com/articles/s41592-019-0638-x>, including library-free mode, where it's highly advantageous. We note that since then DIA-NN has improved significantly in many aspects, including the performance of neural networks, but the general principle of their operation in DIA-NN remains the same.

In this work, we combine all the advantages of DIA-NN, previously demonstrated for regular proteomics (without emphasis on modified peptides), with a new PTM-scoring module (described in detail in the methods section). We show that the resulting workflow substantially outperforms the previous state-of-the-art software for ubiquitinomics (Spectronaut), and this forms part of the impact of our work. In addition to the data processing method, our paper also generates an impact by (i) describing an improved lysis protocol for ubiquitinomics (ii) introducing a new method for ubiquitinome-signature profiling of drug action at high temporal resolution and (iii) using this method to profile USP7 action.

Supplementary Figure 7 has the y-axis value truncated in a way that obscures the true result.

Response 23: Thank you for spotting this. We modified the figure (Supplementary Figure 9 of the revised manuscript).

On p29, the methods describe two different STAGE tip protocols. When were each used? Are there any key performance differences between them that might affect data interpretation?

Response 24: For all generated data in this study we used SDB-RPS desalting of peptides, as described in Methods. The only exception is the urea/SDC comparison of Figure 1. In this case we used C18-desalting of K-GG peptides for urea lysates, as we were following the protocol published by Udeshi et al (doi: 10.1038/nprot.2013.120). Please see also the Methods section (Page 40, starting from line 19).

Reviewer #3 (Remarks to the Author):

The authors describe here a novel workflow for ubiquitomics, the analysis of ubiquitylated proteins in a biological sample. This workflow is based on a sample preparation protocol using an SDC (sodium deoxycholate)-based cell lysis, prior to the enrichment of the modification, followed by data-independent acquisition (DIA) coupled to a neural network-based data processing. This methodology increases significantly the sensitivity and the throughput of the current GlyGly peptidome analysis workflow. Finally, the authors applied this workflow to characterise four different USP7 inhibitors in a colon cancer cell line (HCT116).

General comments:

- The manuscript is well written and clearly presented. However, the use of the English language could be improved.

Thank you! We tried to improve the language and the manuscript was proofread by a native speaker. See changes throughout the manuscript.

- The experiments are well designed and the quality of the figures is good.

Thank you, we are glad to hear this!

- This study is of high interest to other researchers in the field. However, a very similar approach, (SDC-based cell lysis followed by DIA analysis) has been published recently in this journal: Hansen, et al., 2020, <https://doi.org/10.1038/s41467-020-20509-1>. This takes in part the novelty of this study off.

Response 25: We acknowledge that the Mann lab published a study in which it used SDC lysis and DIA-MS for ubiquitinomics earlier this year. However, we also note that we posted our manuscript back-to-back as a preprint on bioRxiv, and this allows us to claim novelty of using DIA for ubiquitinomics.

Nonetheless, we cited and discussed the work by Mann and colleagues in the discussion of the original paper and now included the reference in the first part of the results section (Page 4, line number 9). Although this study is similar topic-wise, our lysis protocol differs from the protocol applied by Hansen et al. Moreover, in the revised version of the manuscript, we demonstrate that DIA-NN significantly outperforms (+ 45%/75% K-GG peptides quantified) the DIA data processing software used by Hansen et al (i.e. Spectronaut, see Figures 2g, 3e and Supplementary Figure 8 of the revised manuscript).

Major points:

- Lack of orthogonal validation of the described workflow using more basic biochemistry techniques to perform ubiquitylation assays. For example, doing GST-UBA pulldowns as in Hansen, et al., 2020 (Fig S3) or performing Ubiquitin immunoprecipitations as in Akimov et al., 2018 (<https://doi.org/10.1038/s41594-018-0084-y>; Figure 5). This would be particularly interesting for validating some of the USP7-dependent identified GG peptides.

Response 26: As suggested by the reviewer, we included such a validation experiment in the revised manuscript. Specifically, we used tandem ubiquitin binding entities (TUBEs) to enrich ubiquitinated proteins after DMSO/FT671 treatment. To detect all enriched proteins in an unbiased manner and to precisely quantify them, we use mass spectrometry as readout. Specifically, we digested all proteins on-bead, followed by MS analysis. This clearly demonstrated that a large fraction of proteins that were strongly ubiquitinated after 15 minutes of FT671 treatment (see Figure 4 in the manuscript) were also significantly enriched in the pulldown experiment using ubiquitin domains (Supplementary Figure 6 of the revised manuscript). These included USP7 and Topors, two of the most strongly regulated proteins with both assays. Additionally, we confirmed USP7 ubiquitination after FT671 treatment by TUBE pulldown and Western blotting (Supplementary Figure 6). For changes in the text, please see Page 14, starting from line 21).

Moreover, for confirming USP7-dependent ubiquitination events, we performed an siRNA-mediated knockdown of USP7, followed by inhibitor treatment and mass spectrometry-based analysis of the ubiquitinome. This experiment clearly confirmed that the observed increase in protein ubiquitination is USP7-dependent. This data is presented in Figure 4 and in Supplementary Figure 5 of the revised manuscript.

- The acquisition methods (LC-MS/MS measurements) are not comparable for DDA (more conservative) and DIA (more relaxed) and this could lead to different sensitivities. For instance, I

consider the average of 4 data points low (8-10 data points would be my recommendation). Table S8 describing the optimisation of the number of DIA segments is missing.

Response 27: First of all, we apologize for the incorrect assignment of table indices. We intended Table S4 (now called Supplementary Data 4) and we corrected this in the revised manuscript (Page 42, lines 7-8).

As for the employed MS methods for DDA and DIA, we used the best performing methods for both acquisition modes. For DDA, this is a Top15 method and for DIA, we found that an average of 4-5 data points per peak was the best compromise between identification numbers and precision of quantification (CVs). As we have shown, our workflow has good precision, significantly better than DDA, in addition to offering several-fold improvement in identification numbers.

We systematically tested this with 7/6/5/4 data points per peak but did not include the data in the manuscript in order to keep the manuscript concise. The parameters of the optimized methods as well as the LC parameters are in Supplementary Data 4.

- ~10% of the GG peptides identified by DDA do not overlap with the GG peptides identified by DIA. How would you explain this if the DIA method is more sensitive?

Response 28: There are multiple possible explanations for this observation. First of all, the standard settings in the two softwares are different. For instance, MaxQuant uses 2 missed cleavages per default while we allowed only one in DIA-NN. As a consequence, some of the peptides will be different when comparing the two software. Second, DIA will fail to identify peptides which are co-fragmented together with some co-eluting peptides that produce the same or very close fragment ion masses. Further, with PTM-enabled searches, any DIA software, be it Spectronaut or DIA-NN, is also forced to discard all peptide-spectrum matches that do not contain enough evidence for the presence (or absence) of the modification(s) in the process of PTM scoring. On the other hand, DDA will fail to identify peptides which are not selected for fragmentation. The ability of DIA to identify substantially more peptides and with higher data completeness comes, among other things, from the fact that it misses less peptides than DDA. However, DIA is not guaranteed to identify all the peptides identified by DDA.

Minor points:

- The authors use USP7 to refer to the human protein. The right nomenclature is USP7 and this should be amended.

Apologies for the confusion. We have corrected it throughout the manuscript.

- The authors mention that simultaneously recording the dynamics of both proteome and ubiquitome is a strategy for capturing USP7 targets or substrates (i.e. in the abstract or on page 11). This is not correct since these techniques do not provide evidence for direct interaction and we cannot rule out that the observed effects might be a consequence of secondary ubiquitylation events. The should be more carefully reformulated across the manuscript.

Response 29: It is true that the observed changes in proteome abundances could be indirect effects and we believe that many of those are indeed indirect, especially at late time points. However, the evidence of increased ubiquitination at early time points together with a downregulation of the protein at intermediate time points (30 min-60 min) represents very strong evidence that these are direct USP7 substrates. We further do not claim that these are bona-fide substrates, as this would need to be followed up, ideally with an enzymatic assay using purified proteins. We state that ‘...*this allows to pinpoint putative substrates with high confidence*’, which we believe is a correct statement.

- deep neural network-based data processing is mentioned several times across the text. A brief description of such an approach would be useful for gaining access to a broader audience.

Response 30: As we have noted above (Response 22), the advantages of the neural network approach implemented in DIA-NN are discussed and thoroughly benchmarked in the original DIA-NN paper (<https://www.nature.com/articles/s41592-019-0638-x>), while here we have combined this neural network approach with a PTM scoring module. We demonstrate that this is a very significant improvement over state-of-the-art for ubiquitinomics (Figures 2, 3 and Supplementary Figure 8). The details of this scoring module are described in detail in the methods section of the revised manuscript.

REVIEWERS' COMMENTS

Reviewer #1 (Remarks to the Author):

The authors have addressed all my concerns satisfactorily, and I look forward to the publication of this manuscript and the related works they mentioned wrt DIA-NN updates.

Reviewer #2 (Remarks to the Author):

The authors are commended for assembling an impressive body of work. Their detailed responses to the original review have greatly improved what was already a terrific body of work. This paper will be impactful for both the MS proteomics and the ubiquitin research communities.

Reviewer #3 (Remarks to the Author):

The authors satisfactorily addressed my main concerns in the revised version of the manuscript and I recommend the publication in its current form.

Just a couple of very minor comments:

- Ubiquitinome is correct but ubiquitome or ubiquitylome are also correct (Personally, I prefer ubiquitylome).

- Even if you show changes in ubiquitylation/protein levels at very short times after treatment with USP7 inhibitors, that does not mean that these proteins are direct substrates.

Ubiquitylation/degradation occurs very quickly and intermediate factors might still be involved.

Additional interactomics, proximity labelling, and biochemical analyses would be required to be fully confident.

Point-by-point response

Summary

We again thank all Reviewers for their great work in assessing our manuscript and for their positive feedback. We respond to the reviewers comments below.

Reviewer #1 (Remarks to the Author):

The authors have addressed all my concerns satisfactorily, and I look forward to the publication of this manuscript and the related works they mentioned wrt DIA-NN updates.

Thank you very much!

Reviewer #2 (Remarks to the Author):

The authors are commended for assembling an impressive body of work. Their detailed responses to the original review have greatly improved what was already a terrific body of work. This paper will be impactful for both the MS proteomics and the ubiquitin research communities.

Thank you, we are delighted to get such a great feedback!

Reviewer #3 (Remarks to the Author):

The authors satisfactorily addressed my main concerns in the revised version of the manuscript and I recommend the publication in its current form.

Thank you!

Just a couple of very minor comments:

- Ubiquitinome is correct but ubiquitome or ubiquitylome are also correct (Personally, I prefer ubiquitylome).

We fully agree that both terms are used. We added one sentence in the introduction making it clear that the two terms are interchangeable (Page 4, third paragraph).

- Even if you show changes in ubiquitylation/protein levels at very short times after treatment with USP7 inhibitors, that does not mean that these proteins are direct substrates. Ubiquitylation/degradation occurs very quickly and intermediate factors might still be involved. Additional interactomics, proximity labelling, and biochemical analyses would be required to be fully confident.

We agree that no standalone assay can interrogate an enzyme-substrate relationship satisfactorily and that orthogonal assays are required for this. We added the following sentence to the discussion (Page 13, first paragraph):

By doing so, early-induced ubiquitination events on a target protein can be connected to its degradation at later time points, allowing the pinpointing of putative substrates with high confidence. Nevertheless, to confirm direct DUB targets, orthogonal assays such in vitro deubiquitination assays or proximity-labeling experiments are required.